# Generation of the organotypic kidney structure by integrating pluripotent stem cell-derived renal stroma

Shunsuke Tanigawa[1,4], Etsuko Tanaka[1,4], Koichiro Miike[1], Tomoko Ohmori[1], Daisuke Inoue[1], Chen-Leng Cai [2], Atsuhiro Taguchi [1,3], Akio Kobayashi[1] & Ryuichi Nishinakamura [1✉]

Organs consist of the parenchyma and stroma, the latter of which coordinates the generation of organotypic structures. Despite recent advances in organoid technology, induction of organ-specific stroma and recapitulation of complex organ configurations from pluripotent stem cells (PSCs) have remained challenging. By elucidating the in vivo molecular features of the renal stromal lineage at a single-cell resolution level, we herein establish an in vitro induction protocol for stromal progenitors (SPs) from mouse PSCs. When the induced SPs are assembled with two differentially induced parenchymal progenitors (nephron progenitors and ureteric buds), the completely PSC-derived organoids reproduce the complex kidney structure, with multiple types of stromal cells distributed along differentiating nephrons and branching ureteric buds. Thus, integration of PSC-derived lineage-specific stroma into parenchymal organoids will pave the way toward recapitulation of the organotypic architecture and functions.

[1] Department of Kidney Development, Institute of Molecular Embryology and Genetics, Kumamoto University, Kumamoto 860-0811, Japan. [2] Department of Pediatrics, Indiana University School of Medicine, Indianapolis, IN 46202, USA. [3] Present address: Department of Genome Regulation, Max Planck Institute for Molecular Genetics, Berlin, Germany. [4] These authors contributed equally: Shunsuke Tanigawa and Etsuko Tanaka. ✉email: ryuichi@kumamoto-u.ac.jp

Recent progress in stem cell biology has enabled the generation of "organoids" from pluripotent stem cells (PSCs), but most of the currently available organoids lack the organotypic "higher-order structure", in which functional units are distributed at the periphery of a branching epithelial backbone. One of the essential pieces for this organization is the stroma, which develops with epithelial parenchymal cells. However, previous studies mainly focused on generating the parenchyma and paid little attention to the authenticity of the stroma co-induced in the organoids[1]. Because it has become apparent that the stroma of individual organs exhibits distinct characteristics and gene expression patterns[2–4], we focused on the development of organ-specific stroma.

The embryonic kidney, the metanephros, forms its complex structure by mutual interactions of the metanephric mesenchyme (MM; including nephron progenitors [NPs] and stromal progenitors [SPs]) and the ureteric bud (UB)[5–8]. NPs and UBs are the parenchymal precursors. NPs form the capping mesenchyme around the UB tips and secrete GDNF to induce continuous branching of UBs that eventually develop into the urine-collecting system: the collecting ducts and ureter. In turn, UBs emit WNT and FGF signals that maintain NPs while inducing a subset of NPs to differentiate into nephrons composed of glomeruli and renal tubules[9,10]. Meanwhile, SPs located in the periphery of the metanephros produce GDNF and ALDH1A2, the latter of which maintains GDNF receptor Ret expression in UB tips via retinoic acid (RA) signaling to support UB branching[11]. SPs also express FAT4 to suppress excessive NP proliferation[12], allowing balanced nephron differentiation. In addition to their roles in UB branching and NP differentiation, SPs themselves differentiate into multiple types of interstitial cells. Lineage-tracing experiments showed that SPs expressing the transcription factor FOXD1 differentiate into the majority of the intrarenal stroma (interstitial cells), which is organized in a corticomedullary manner and surrounds the NP- and UB-derived epithelial parenchyma[3,7]. SPs also differentiate into functionally specialized stromal cells such as mesangial cells and renin cells, which regulate glomerular filtration and systemic blood pressure, respectively[7,13]. Thus, this triad (NP-UB-SP) interaction is essential for the formation of the organotypic "higher-order structure" of the kidney: numerous nephrons distributed at the periphery of branching UB/collecting ducts, all surrounded by multiple types of stromal cells.

We and others reported the induction of kidney organoids from mouse and human PSCs[14–16]. Most of these studies involved induction of NPs, resulting in nephron formation in the organoids (NP organoids). We subsequently reported a protocol for selective induction of UBs, which show branching morphogenesis in gels (UB organoids), from mouse embryonic stem cells (ESCs) and human induced pluripotent stem cells (iPSCs)[17]. Notably, when NPs and UBs differentially induced from mouse ESCs were combined with stromal cells isolated from mouse embryonic kidneys, the organotypic "higher-order structure" was reproduced, namely the peripheral progenitor niche and internally differentiated nephrons interconnected by a dichotomously branching ureteric epithelium. However, this structure was not formed in the absence of stromal cells[17], as expected from the accumulated knowledge in vivo[12,18]. Recently, several groups generated human iPSC-derived UB organoids and combined them with NP organoids[19–21], but proper UB branching or differentiation of kidney-specific stromal cell types was not observed. Although all conventional organoids contain non-parenchymal "stroma-like" cells, their identity and differences from the in vivo counterparts remain obscure. Thus, the establishment of protocols for SP induction from PSCs precisely following the in vivo developmental trajectory is essential to

recapitulate the generation of kidney-specific stromal cells and the organotypic architecture.

We previously identified spatiotemporally distinct origins for the MM (NPs + SPs) and the UB in mice[14], which enabled us to establish induction protocols for NP and UB organoids[14,17]. The $T^+$ immature mesoderm at embryonic day (E) 7.5 turns into the $Osr1^+$ anterior intermediate mesoderm (IM) at E8.5, and eventually differentiates into the UB. Meanwhile, the MM precursor is maintained in the caudal $T^+$ immature state for a longer period up to E8.5, and then differentiates into the $Osr1^+$ posterior IM at E9.5. As the $Osr1^+$ posterior IM further develops into $Six2^+$ NPs and $Foxd1^+$ SPs at E11.5[14,22], we hypothesized that modification of our NP induction protocol after the E9.5 posterior IM stage would lead to the establishment of SP induction and eventually generate the "higher-order structure" of the kidney.

In this work, we use the reverse induction approach established in our previous NP and UB lineage induction studies[14,17]. Briefly, we initially focus on the last step of SP differentiation (from E9.5 $Osr1^+$ posterior IM to E11.5 $Foxd1^+$ SPs) and then combine the protocol with that for the earlier stages (from mouse ESCs to posterior IM). Finally, the obtained ESC-derived SPs are combined with ESC-derived NPs and UBs to show the generation of the organotypic kidney structure, as well as the differentiation into kidney-specific stromal cell types.

## Results

### Contribution of dorsal SPs to UB branching and renal stroma.

In a previous study[17], we showed that combining E11.5 embryonic kidney-derived PDGFRA$^+$ stromal cells with NPs and an isolated UB reconstituted the well-branched collecting duct structure in vitro. Because the $Foxd1^+$ subpopulation among PDGFRA$^+$ stromal cells gives rise to most of the kidney stroma[7], we sorted GFP$^+$PDGFRA$^+$ SPs from E11.5 $Foxd1$-$GFP$ mice and aggregated them with NPs and UBs (Supplementary Fig. 1a, b). The results showed that $Foxd1^+$ SPs induced UB branching more efficiently than $Foxd1^-$ cells (Supplementary Fig. 1c), proving that inclusion of $Foxd1^+$ SPs was essential for the generation of the organotypic structure. Immunostaining of the E11.5 kidney revealed that stromal domains expressing FOXD1, TBX18, and ISL1 were aligned along the dorsoventral axis, with the FOXD1$^+$ domain located most dorsally and adjacent to SIX2$^+$ NPs (Fig. 1a). We genetically labeled the three stromal domains at this stage using $Foxd1CreER^{T2}$, $Tbx18MerCreMer$, and $Isl1MerCreMer$ mice, and confirmed that $Foxd1^+$ SPs contributed to the majority of the intrarenal stroma at E15.5, $Tbx18^+$ SPs mainly contributed to the stroma surrounding the ureter, and $Isl1^+$ SPs contributed to the lower urinary tract including the bladder and urethra (Supplementary Fig. 1d), consistent with the previous reports[7,23,24].

### Dorsoventral SP patterning by FGF/BMP signaling.

To understand the development of SPs along the dorsoventral axis, we performed single-cell RNA-sequencing (scRNA-seq) of the E11.5 kidney, which revealed separate clusters of NPs ($Six2^+$), UBs ($Ret^+$), stromal cells ($Pdgfra^+$), and endothelial cells (ECs; $Pecam1^+$) in UMAP plots (Supplementary Fig. 1e). Re-clustering of the extracted stromal cells showed that the stromal cells could be further divided into five subdomains: two with proliferation-related signatures and three likely representing $Foxd1^+$, $Tbx18^+$, and $Isl1^+$ progenitors (Fig. 1b, c). Marker genes for the three populations were also identified (Fig. 1c, d). Several signaling-related genes were expressed in the stroma: $Rarb$ (RA signal indicator) in the entire stroma, $Etv4$ and $Etv5$ (FGF signal indicators) in the $Foxd1^+$ domain, and $Id1$ and $Smad7$ (BMP signal indicators) in the $Tbx18^+$ and $Isl1^+$ domains (Fig. 1e). The

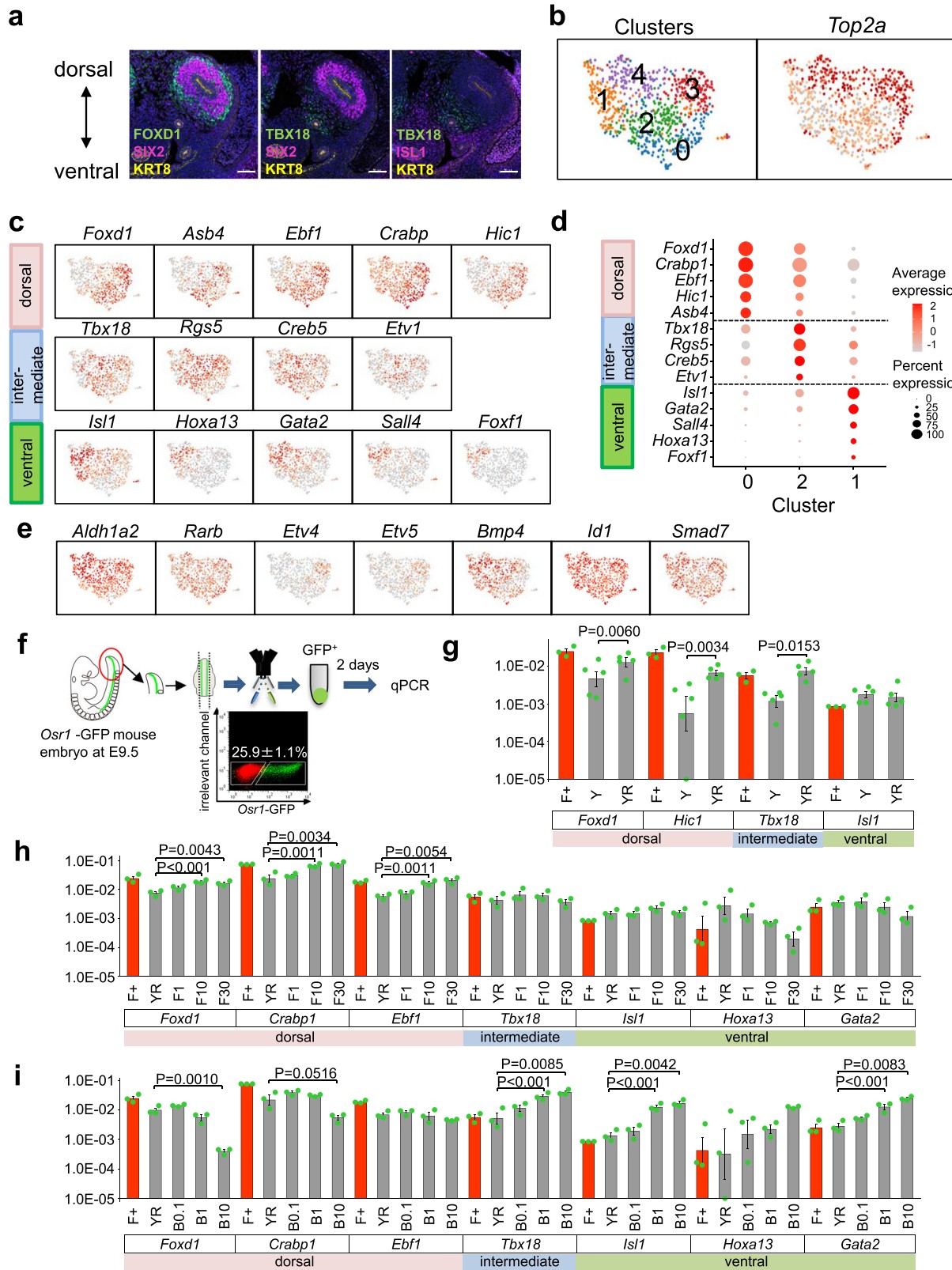

expression of signaling ligands/synthesis enzymes (Fig. 1e and Supplementary Fig. 1e) was consistent with previous in vivo studies, including *FGF20* expression in NPs, *FGF9* in NPs and UB tips[10], *BMP4* in the ventral stroma adjacent to the Wolffian duct[25], and *Aldh1a2* in the cortical stroma[11]. Thus, we hypothesized that dorsoventral signal gradients may exist in the

developing renal stroma: higher FGF signal dorsally, higher BMP signal ventrally, and ubiquitous RA signal throughout the stroma.

A previous cell lineage study demonstrated that NPs and SPs are derived from *Osr1+* cells, present in the IM and part of the lateral plate mesoderm (LPM) at E9.5[22]. To examine the FGF- and BMP-driven dorsoventral patterning of the stroma, we

**Fig. 1 RA, FGF, and BMP signaling regulate dorsoventral patterning of renal SPs. a** Three domains of SPs, shown by immunostaining of the kidney at E11.5. Two biologically independent mice were examined in two separate experiments. Left panel: FOXD1+ dorsal SPs (green); middle panel: TBX18+ intermediate SPs (green); right panel: ISL1+ ventral SPs (magenta). SIX2 and KRT8 are also stained to mark NPs and UBs, respectively. Scale bars: 15 μm. **b–e** scRNA-seq analysis of the stromal cells in the mouse E11.5 kidney. **b** UMAP plots showing five clusters, with two representing *Top2a*+ proliferating cells. Representative genes in dorsal, intermediate, and ventral SPs, shown as UMAP plots (**c**) and dot plots (**d**). **e** UMAP plots showing signal-related genes. **f** Schematic diagram of isolation of *Osr1-GFP*+ cells from E9.5 embryos, followed by a 2-day culture. **g–i** Expression of stromal domain-related genes after the culture, analyzed by qRT-PCR. **g** RA induces dorsal SP-related genes (*n* = 5 biologically independent experiments). **h, i** RA and FGF9 induce dorsal SP-related genes, while RA and BMP4 induce intermediate and ventral SP-related genes (*n* = 3 biologically independent experiments). Relative mRNA expression levels normalized to β-actin gene expression are shown as mean ± SEM. Two-sided Student's *t*-test was performed in (**g**) and Dunnett's multiple comparison test (two-sided) was performed in (**h, i**). The source data are provided as a Source Data file. F+: Foxd1-GFP+PDGFRA+ dorsal SPs harvested at E11.5 (non-cultured; presented as a reference); Y: Y27632 (10 μM); R: RA (0.1 μM); F1: FGF9 (1 ng/ml); F10: FGF9 (10 ng/ml); F30: FGF9 (30 ng/ml); B0.1: BMP4 (0.1 ng/ml); B1: BMP4 (1 ng/ml); B10: BMP4 (10 ng/ml).

surgically removed the LPM and hindlimb buds from *Osr1-GFP* mice at E9.5, sorted the GFP+ posterior IM cells, and cultured them as spheres for 2 days in the presence of the above-mentioned factors, as well as Y27632 to support cell survival (Fig. 1f). Treatment of GFP+ posterior IM cells with FGF9 and a Wnt agonist CHIR99021 (CHIR) was previously shown to result in NP generation, while the addition of RA inhibited the NP induction[14]. In the present study, RA treatment of the same population enhanced two dorsal SP markers, *Foxd1* and *Hic1*, in addition to general stromal markers *Pbx1* and *Fat4* (Fig. 1g and Supplementary Fig. 1f). The addition of FGF9 to Y27632 and RA (YRF) further increased dorsal SP markers (*Foxd1*, *Crabp1*, *Ebf1*, *Asb4*), while ventral markers (*Hoxa13*, *Gata2*) were mildly decreased (Fig. 1h and Supplementary Fig. 1g). In contrast, BMP4 addition (YRB) markedly reduced dorsal SP markers (*Foxd1*, *Crabp1*), while markers for intermediate SPs (*Tbx18*, *Creb5*) and ventral SPs (*Isl1*, *Gata2*) were upregulated (Fig. 1i and Supplementary Fig. 1h). Thus, two signaling pathways are likely to regulate the dorsoventral patterning of renal SPs: FGF9 induces dorsal SPs, while BMP4 induces ventral and intermediate SPs.

**Isolation of the posterior IM by ROBO2-based sorting**. To further evaluate the differentiation efficiency from the posterior IM toward the dorsoventral SPs, we aimed to sort the posterior IM from *Foxd1-GFP* mouse embryos. Because *Foxd1-GFP* is not expressed in the posterior IM population at E9.5, we searched for cell surface markers of the posterior IM. An scRNA-seq analysis of the posterior part (caudal from the 26th somite) of E9.5 embryos showed that the *T*+ tail bud mesenchyme segregated into neurons and somites (paraxial mesoderm [PM]). The LPM bifurcated into splanchnic (visceral) mesoderm and somatic (parietal) mesoderm (Fig. 2a and Supplementary Fig. 2a). *Osr1* was strongly expressed in the two clusters (IM and LPM), and weakly detectable in some dispersed cells adjacent to the pre-somitic mesoderm (PSM) (Fig. 2b). *Wt1* showed similar, but more restricted, expression patterns. *Grem1*, known to be expressed in the IM and MM[26,27], was detected in the IM cluster, but not in the LPM. *Six2* expression was more restricted to a portion of the IM cluster. We noted that the expression of transmembrane receptor *Robo2* mostly overlapped with the *Osr1*+ IM and LPM domains (Fig. 2b). In situ hybridization on a posterior region similar to that used for the scRNA-seq analysis confirmed that *Osr1* and *Robo2* mostly overlapped in the IM and LPM (Fig. 2c and Supplementary Fig. 2b). *Grem1* was specifically expressed in the posterior IM that co-expressed *Osr1* and *Robo2* (Fig. 2c and Supplementary Fig. 2b).

UMAP plots showed that *Robo2*-positive neural tubes were negative for *Pdgfra*, while *Pdgfra* was expressed in the posterior IM, LPM, and PSM (Fig. 2b). Thus, we decided to test the combination of ROBO2 and PDGFRA for sorting the posterior IM. Because *Osr1* and *Robo2* were also expressed in the LPM, we surgically

removed the LPM and hindlimb buds from the posterior part of E9.5 *Osr1-GFP* mice, and analyzed ROBO2/PDGFRA expression by flow cytometry (Fig. 2d). The ROBO2+PDGFRA+ fraction largely consisted of *Osr1-GFP*high cells, while the ROBO2−PDGFRA+ fraction exhibited varying degrees of *Osr1-GFP* expression (Fig. 2e and Supplementary Fig. 2c). The expression levels of IM marker genes (*Osr1*, *Grem1*, *Wt1*) were high in the ROBO2+PDGFRA+ fraction (Fig. 2f), while PSM-related genes were enriched in the ROBO2−PDGFRA+ fraction (Supplementary Fig. 2d). Thus, the method allowed successful isolation of the posterior IM without genetic fluorescent labeling.

**Induction of dorsal SPs from the embryonic posterior IM**. We sorted the ROBO2+PDGFRA+ posterior IM and ROBO2−PDGFRA+ non-IM fractions from wild-type mice and cultured them in the presence of RA and FGF9 (YRF), based on the results shown in Fig. 1. After 2 days of culture, the sizes of the spheres from the ROBO2+PDGFRA+ fraction and *Osr1-GFP*+ posterior IM fraction were significantly larger than those from the ROBO2−PDGFRA+ fraction (Fig. 2g, upper panels and right graph). A discrete PDGFRA+ putative SP population was generated from the ROBO2+PDGFRA+ fraction, as well as from the *Osr1-GFP*+ posterior IM fraction, but not from the ROBO2−PDGFRA+ non-IM fraction (Fig. 2g, lower panels). We then examined the gene expressions in the induced PDGFRA+ putative SPs. The ROBO2+PDGFRA+ IM-derived cells showed higher expression levels of pan-stromal genes and dorsal SP genes, as well as intermediate and ventral SP genes, than the ROBO2−PDGFRA+ non-IM-derived cells (Fig. 2h and Supplementary Fig. S2e), and their expression levels were almost comparable to those in cultured *Osr1-GFP*+ IM cells and freshly isolated E11.5 dorsal SPs in vivo. Furthermore, when the PDGFRA+ putative SPs derived from the ROBO2+PDGFRA+ IM fraction were combined with E11.5 NPs and UBs, robust UB branching was observed (Fig. 2i and Supplementary Fig. 2f). Similar experiments were not achievable from the ROBO2−PDGFRA+ fraction because few PDGFRA+ cells were obtained after culture. Therefore, the ROBO2+PDGFRA+ posterior IM can be induced to renal dorsal SP-like cells in vitro.

We examined the actual proportion of *Foxd1*+ cells induced in our culture conditions by utilizing *Foxd1-GFP* embryos. However, the YRF culture medium induced minimal *Foxd1*+ cells after 2 days of culture (Supplementary Fig. 2h). It is well known that SHH expressed in the notochord medializes the embryonic mesoderm, thereby patterning the mesoderm in a mediolateral direction[28,29]. In contrast, BMP4 expressed in the LPM antagonizes the SHH-mediated mediolateral patterning of the mesoderm[30,31]. In addition, *Grem1*, a BMP antagonist, was expressed in the posterior IM at E9.5, and BMP activity was lower in the *Foxd1*+ dorsal SPs at E11.5, as shown in Fig. 1. Indeed, the addition of SHH (S) to the YRF culture media increased the

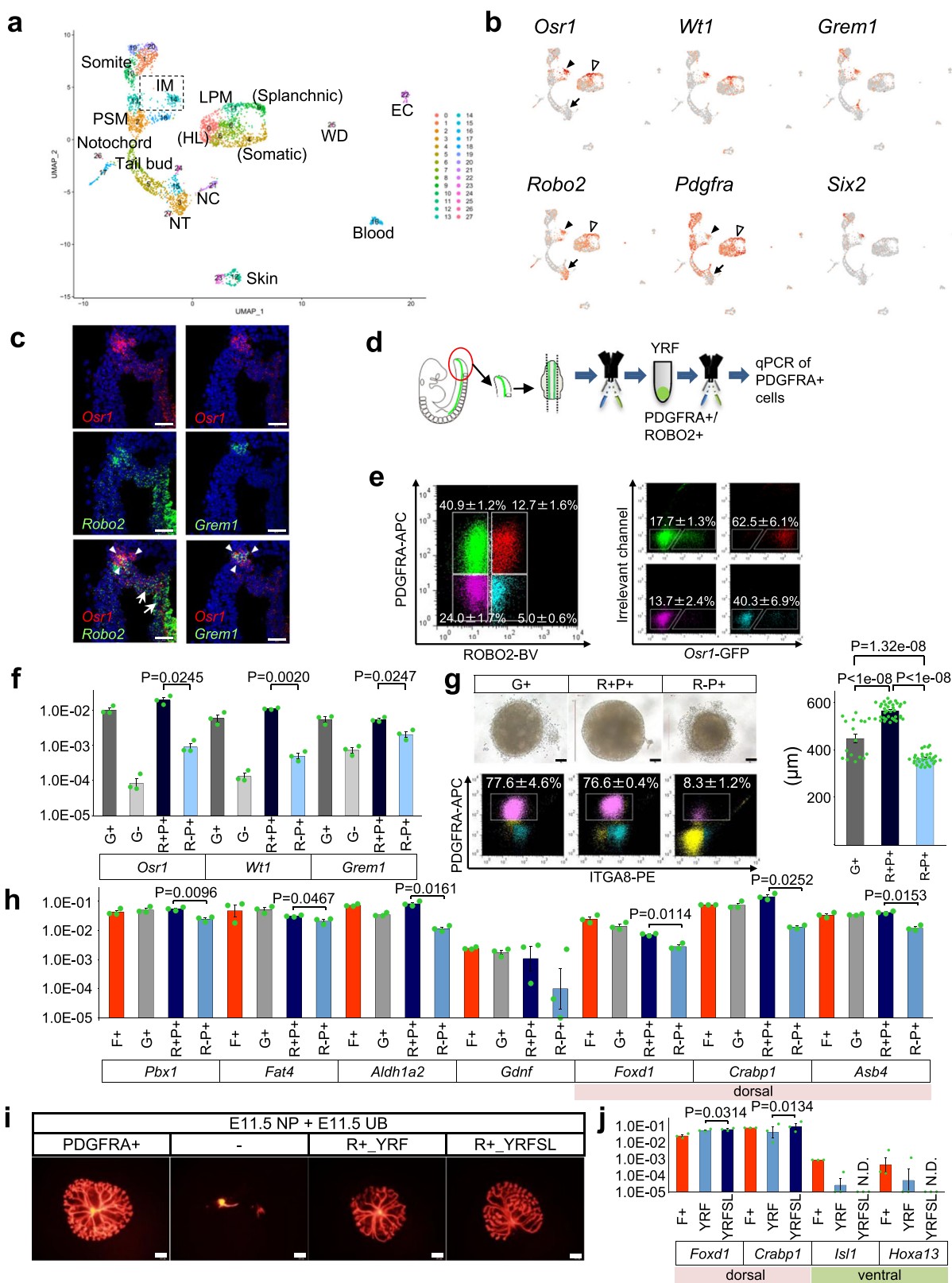

sphere sizes and percentages of PDGFRA$^+$ cells in the spheres (Supplementary Fig. 2g). When we further added the BMP receptor antagonist LDN193189 (L) to culture of the ROBO2$^+$PDGFRA$^+$ fraction from E9.5 *Foxd1-GFP* embryos (YRFSL condition), higher percentages of GFP$^+$ cells were detected after the culture, compared with the YRF condition

(Supplementary Fig. 2h). Indeed, dorsal SP genes were increased, while ventral SP genes were downregulated (Fig. 2j and Supplementary Fig. 2i). The functional competence of the cultured cells was confirmed by aggregation with E11.5 embryonic kidney-derived NPs and an isolated UB, which resulted in extensive UB branching (Fig. 2i and Supplementary

**Fig. 2 ROBO2$^+$ PDGFRA$^+$ IM is induced toward dorsal SPs in vitro. a, b** scRNA-seq analysis of the posterior part of the E9.5 mouse embryo. **a** UMAP plots. **b** Representative genes for the IM. Black arrowheads: IM; white arrowheads: LPM; arrows: neural tube (NT). NC: neural crest; HL: hindlimb bud; WD: Wolffian duct; EC: endothelial cell. **c** In situ hybridization of *Osr1*, *Robo2*, and *Grem1* in the posterior IM at E9.5. Two biologically independent mice were examined in two separate experiments. Arrowheads: IM; arrows: LPM. Scale bars: 50 μm. **d** Isolation of ROBO2$^+$PDGFRA$^+$ cells from wild-type or *Osr1-GFP* embryos at E9.5, followed by a 2-day culture under the YRF condition (Y: Y27632; R: RA; F: FGF9). **e** Flow cytometric analysis before the culture. **f** Expression of IM-related genes before the culture. G+: *Osr1-GFP*$^+$ cells; G−: *Osr1-GFP*$^−$cells; R−P+: ROBO2$^−$PDGFRA$^+$ cells; R+P+: ROBO2$^+$PDGFRA$^+$ cells. Data are shown as mean ± SEM (*n* = 3 biologically independent experiments). Two-sided Student's *t*-test was performed. **g** Spheres after the culture. Upper panels: Spheres from the indicated fractions. Scale bars: 100 μm. Right graph: Diameters of the spheres. Data are shown as mean ± SEM (*n* = 17, 36, and 31 biologically independent samples, respectively). The Tukey–Kramer test (two-sided) was performed. Lower panels: flowcytometric analysis of the induced spheres. **h** Expression of stromal domain-related genes after the culture. F+: Foxd1-GFP$^+$PDGFRA$^+$ dorsal SPs harvested at E11.5 (presented as a reference). G+: *Osr1-GFP*$^+$ cells; R−P+: ROBO2$^−$PDGFRA$^+$ cells; R+P+: ROBO2$^+$PDGFRA$^+$ cells. Data are shown as mean ± SEM (*n* = 3 biologically independent experiments). Two-sided Student's *t*-test was performed. **i** Aggregation assay for UB branching. The ROBO2$^+$PDGFRA$^+$ (R+) fractions were cultured for 2 days in the indicated conditions, combined with E11.5 embryo-derived NPs and UBs (tdTomato$^+$). Scale bars: 200 μm. PDGFRA$^+$: freshly isolated stromal cells from E11.5 embryos used as a reference for re-aggregation. **j** Expression of genes related to dorsoventral patterning in GFP$^+$ cells cultured from the ROBO2$^+$ PDGFRA$^+$ fraction of E9.5 *Foxd1-GFP* embryos. Data are shown as mean ± SEM (*n* = 3 biologically independent experiments). Two-sided Student's *t*-test was performed. **f**–**h**, **j** The source data are provided as a Source Data file.

Fig. 2f). Thus, the YRFSL condition is likely to represent the optimal condition for induction of dorsal SPs from the posterior IM in embryos.

**Induction of dorsal SPs from mouse ESCs.** Next, we examined whether mouse ESCs can be directionally differentiated into SPs. We adopted the previously reported NP induction protocol up to day 6.5[14], when the cell population is likely equivalent to the posterior IM (Fig. 3a). At this stage, most of the induced cells were positive for ROBO2 (Fig. 3b). We sorted the ROBO2$^+$PDGFRA$^+$ fraction and cultured the cells in the SP induction condition established above. Similar to the case for cells from embryos, treatment with FGF9 (YRF) retained dorsal SP genes, while BMP4 (YRB) increased intermediate SP genes (Supplementary Fig. 3a). However, the YRF condition alone exhibited poor growth of spheres, and the addition of SHH to the culture led to marked increases in viability and PDGFRA$^+$ cell numbers (Supplementary Fig. 3b). We further added LDN193189 and found that expression of dorsal SP genes was increased, while expression of intermediate and ventral SP genes was decreased (Supplementary Fig. 3c). Furthermore, we compared the induced SPs (iSPs) with non-directionally induced stroma (nd-iS, Supplementary Fig. 3d): PDGFRA$^+$ cells co-induced from mouse ESCs by the NP induction protocol, which employed Y27632, FGF9, and CHIR in the last step[14]. As expected, the expression of dorsal SP genes was higher in iSPs than in nd-iS (Fig. 3c). *Aldh1a2*, which regulates UB branching, and *Fat4*, which controls NP differentiation, were expressed more abundantly in iSPs than in nd-iS (Fig. 3c). This induction protocol was also applicable to another mouse ESC line (G4-2), in which the iSPs exhibited similar gene expression features (Supplementary Fig. 3e).

scRNA-seq analyses of the iSPs showed the main *Pdgfra*$^+$ iSP clusters, with a few off-target clusters that were negative for *Pdgfra* but expressed genes related to ECs, neurons, or muscles (Fig. 3d). The NP induction protocol from mouse ESCs produced ITGA8$^+$PDGFRA$^−$ NP clusters, PDGFRA$^+$ nd-iS clusters, and a few PDGFRA$^−$ (EC-like, neuron-like, or muscle-like) off-target clusters (Fig. 3d). The merged UMAP plots showed that the iSP clusters partly overlapped with the dorsal SPs in the embryonic kidney (Fig. 3d).

We next examined whether the ESC-derived iSPs satisfied the function of SPs; ability to induce UB branching and NP differentiation. For this purpose, we isolated NPs from wild-type E11.5 embryos and UBs from E11.5 *Hoxb7-GFP* embryos, and combined them with the sorted iSPs derived from ESCs (Supplementary Fig. 3f). Reaggregation of NPs and UBs without stromal cells or with nd-iS resulted in poor branching at day 7 of

culture (Fig. 3e, f). In contrast, the inclusion of iSPs produced marked UB branching (Fig. 3e, f). We also confirmed that our optimized condition (YRFSL) evoked greater numbers of UB branches than the condition lacking LDN193189 (Supplementary Fig. 3g). Section in situ hybridization revealed expression of *Ret* in UB tips and *Wnt7b* in UB stalks of the kidney organoids containing iSPs (Fig. 3f, 2nd column), indicating that tip-stalk identity along the corticomedullary axis was established. *Six2*$^+$ NPs and *Foxd1*$^+$ SPs accumulated around *Ret*$^+$ UB tips (Fig. 3f, 3rd column), demonstrating maintenance of NPs and SPs resulting in the formation of the nephrogenic niche. Furthermore, multiple nephron cell types (glomeruli, proximal, and distal renal tubules) were formed in iSP-containing organoids (Fig. 3f, 4th and 5th columns). Thus, mouse ESC-derived iSPs possess the functionality to generate the complex kidney structure when combined with mouse embryo-derived UBs and NPs, by organizing UB branching and NP differentiation.

**Kidney organoids completely derived from ESCs.** We previously reported induction protocols for NP and UB organoids[14,17]. Our scRNA-analysis showed that the mouse ESC-derived iNP and iUB clusters overlapped with their in vivo counterparts (Fig. 3d). Thus, we generated the kidney structure solely derived from ESCs, by combining differentially induced NPs, UBs, and SPs (iNPs, iUBs, and iSPs). We used *Hoxb7-GFP* ESCs[17] that specifically express GFP in the UB lineage to allow monitoring of UB branching in the organoids. We induced the *Hoxb7-GFP* ESCs into the three lineages using the different protocols, reaggregated the three progenitor populations, and cultured them for 7 days in the absence of growth factors (Fig. 4a). PDGFRA$^−$ off-target cells co-induced with iNPs or iSPs, as well as PDGFRA$^+$ nd-iS, were likely to be eliminated by sorting based on ITGA8/PDGFRA expression. The iUBs also contained stroma-like cells (Fig. 3d). However, most of these cells, which were located in the central portion of the organoids[32], were likely to be surgically removed when the aggregates were prepared because we only used epithelial buds sprouting out of the UB organoids.

Organoids derived from the three induced progenitors exhibited robust UB branching, with higher branching numbers than organoids generated from iNPs and iUBs alone, or iNPs, iUBs, and nd-iS (Fig. 4b, c). NPs were distributed at the periphery of the branching UBs (Fig. 4c, 2nd column, Fig. 4d), while nephrons (glomeruli, proximal and distal renal tubules) were formed inside the organoids (Fig. 4c, 3rd column, Fig. 4d). The tip-stalk identity of the UBs was established, as shown by *Ret* and *Wnt7b* expression (Fig. 4c, 4th column), and *Six2*$^+$ NPs and

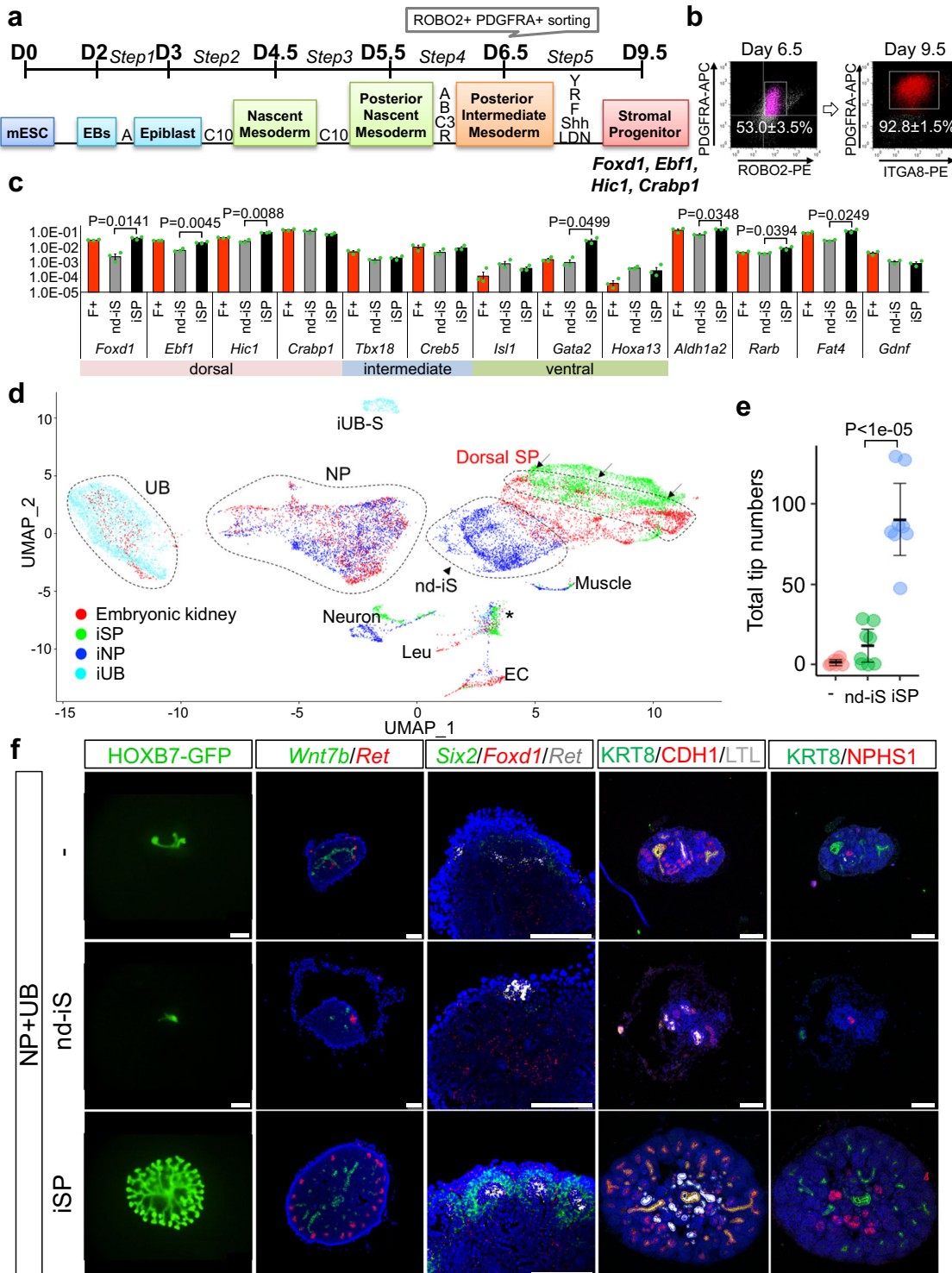

**Fig. 3 Mouse ESC-derived SPs support the generation of the complex kidney structure. a** SP induction protocol from mouse ESCs. A: ActivinA (10 ng/ml); B: BMP4 (3 ng/ml); C10: CHIR (10 μM); C3: CHIR (3 μM); R: RA (0.1 μM); F: FGF9 (10 ng/ml). **b** Flow cytometric analysis of the induced cells at day 6.5 and 9.5. **c** Comparison of expression levels of stromal domain-related genes between nd-iS and iSPs. F+: Foxd1-GFP⁺PDGFRA⁺ dorsal SPs harvested at E11.5 (non-cultured; presented as a reference). Data are shown as mean ± SEM ($n = 3$ biologically independent experiments). Two-sided Student's $t$-test was performed. The source data are provided as a Source Data file. **d** scRNA-seq analysis of mouse ESC-derived progenitors (iSPs, iNPs, iUBs) and E13.5 embryonic kidney. UMAP plots are shown. Arrows: dorsal SPs of the embryonic kidney. Arrowhead: nd-iS co-induced with iNP. EC: endothelial cells; Leu: leucocytes; iUB-S: stroma co-induced with iUB; *: dying cells. **e** UB branch numbers in the aggregates. Data are shown as mean ± SEM ($n = 7$, 7, and 8 biologically independent samples, respectively). The Tukey–Kramer test (two-sided) was performed. The source data are provided as a Source Data file. **f** Aggregates formed without stromal cells (−), with nd-iS, or with iSPs. NPs and UBs are isolated from the E11.5 embryonic kidneys. 1st column: GFP images of UBs; 2nd column: in situ hybridization of *Wnt7* (UB stalks) and *Ret* (UB tips); 3rd column: in situ hybridization of *Six2* (NPs), *Foxd1* (SPs), and *Ret* (UB tips); 4th column: immunostaining of KRT8 (UBs), CDH1 (UBs, distal tubules), and LTL (proximal tubules); 5th column: immunostaining of KRT8 (UBs) and NPHS1 (glomerular podocytes). Scale bars: 100 μm. Six aggregates in each condition obtained from three independent experiments were analyzed.

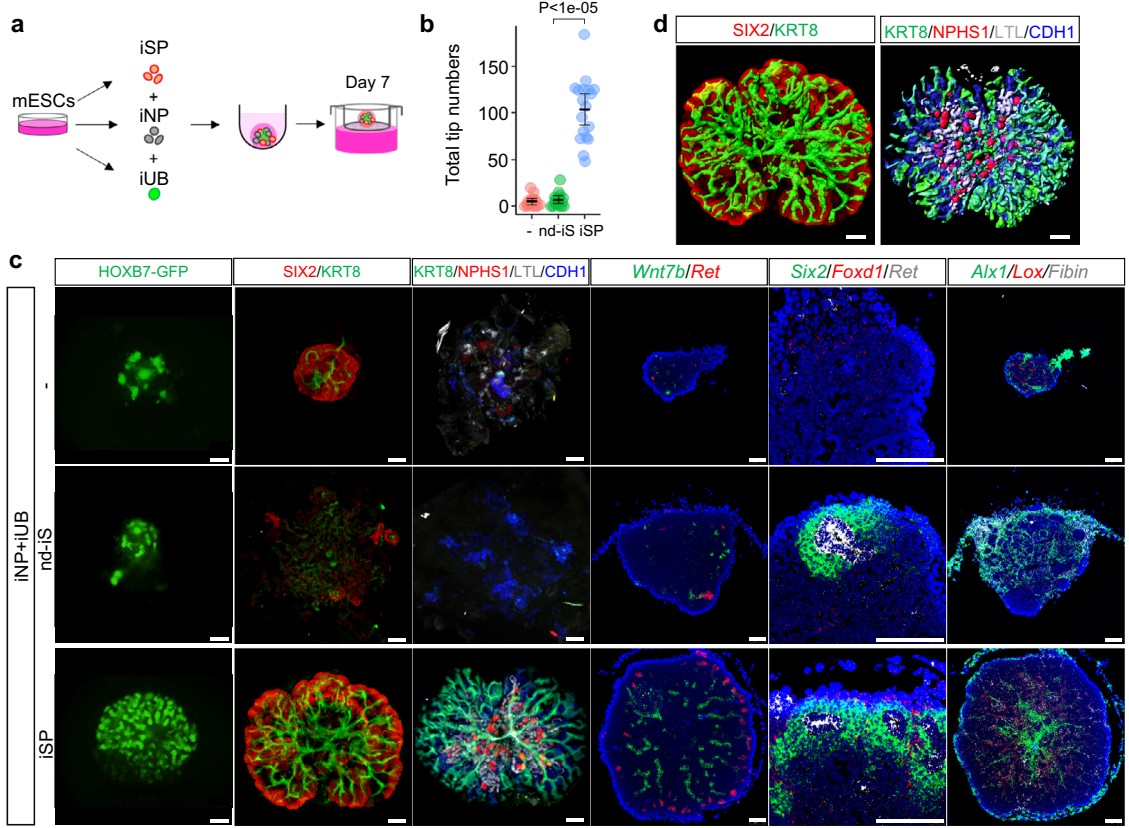

**Fig. 4 Kidney organoids solely derived from ESCs exhibit the organotypic "higher-order structure" in vitro. a** Schematic diagram of kidney organoid formation. iSPs, iNPs, and iUBs derived from *Hoxb7-GFP* mouse ESCs are combined, and cultured at the air/liquid interface for 7 days. **b** UB branch numbers in kidney organoids. Data are shown as mean ± SEM (*n* = 16, 15, and 18 biologically independent samples, respectively). The Tukey–Kramer (two-sided) test was performed. The source data are provided as a Source Data file. **c** Mouse ESC-derived organoids formed without stromal cells (−), with nd-iS, or with iSPs. 1st column: GFP images of UBs; 2nd column: whole-mount immunostaining of SIX2 (NPs) and KRT8 (UBs); 3rd column: whole-mount immunostaining of KRT8 (UBs), NPHS1 (glomerular podocytes), LTL (proximal tubules), and CDH1 (UBs, distal tubules); 4th column: in situ hybridization of *Wnt7* (UB stalks) and *Ret* (UB tips); 5th column: in situ hybridization of *Six2* (NPs), *Foxd1* (SPs), and *Ret* (UB tips); 6th column: in situ hybridization of *Alx1* (medullary stroma), *Lox* (cortical stroma), and *Fibin* (outer layer of cortical stroma). Scale bars: 100 μm. Six organoids in each condition obtained from three independent experiments were analyzed. **d** Digitized images of the whole-mount staining data in the 2nd and 3rd columns of (**c**). Scale bars: 100 μm. Four organoids obtained from two independent experiments were analyzed.

*Foxd1*⁺ SPs accumulated around the UB tips (Fig. 4c, 5th column). The stroma also became patterned in a corticomedullary manner, as determined by in situ hybridization (Fig. 4c, 6th column). A medullary stromal gene *Alx1* was clearly detected in the central region, while a cortical stromal gene *Lox* was expressed in the outer layer of the organoids. Another cortical stromal gene, *Fibin*, was expressed, albeit weakly, in the outermost layer. These expression patterns were similar to those in the embryonic kidney (Supplementary Fig. 4a) and consistent with previous scRNA-seq analyses[2,3]. Thus, by establishing the iSP induction protocol, we successfully generated kidney organoids completely derived from ESCs with recapitulation of the "higher-order structure": nephrons and stroma distributed along the tip-stalk orientation of branching UBs.

**Multiple stromal cell types formed upon transplantation**. To further promote the organoid development, we transplanted the ESC-derived kidney rudiments under the kidney capsule of immunodeficient mice. The kidney organoids with iSPs showed robust UB branching, and had grown in size (1.62 ± 0.25 mm × 0.9 ± 0.18 mm, *n* = 6 biologically independent samples) at day 14 post-transplantation (Fig. 5a). In contrast, the organoids with nd-iS showed poor UB branching (Fig. 5a) and smaller sizes

(1.14 ± 0.26 mm × 0.58 ± 0.15 mm, *n* = 6 biologically independent samples, *p* = 0.003). In this in vivo setting, iUBs underwent further maturation toward collecting ducts with an expression of AQP2 (principal cell marker) and CAR2 (intercalated cell marker) (Fig. 5a, 2nd and 3rd columns). Nephrons were aligned in a corticomedullary manner with peripherally located proximal renal tubules and the centrally located loops of Henle (Fig. 5a, fourth column). A medullary stromal gene *Alx1* was expressed in central regions of the organoids, and the innermost region expressed *Wnt4* in a partly overlapping manner with *Alx1* (Fig. 5b, 1st and 2nd columns), which was similar to the situation in vivo (Supplementary Fig. 4a). These observations indicate the formation of the deeper medullary stroma, and are consistent with a previous report on *Wnt4* expression[33]. In contrast, organoids with nd-iS exhibited poorly patterned nephrons, UBs/collecting ducts, and stroma (Fig. 5a, b).

We examined the development of glomeruli in more detail because SPs in vivo are known to differentiate into specialized stromal cells: mesangial cells, which form inside the glomeruli, and renin cells, which surround the afferent arterioles coming into the glomeruli[7,34]. As we previously reported[35], glomeruli in the transplants were well vascularized with PECAM1⁺ ECs and the slit diaphragm-related protein NEPHRIN was distributed in a linear manner on the basal side (facing the ECs) of the podocytes

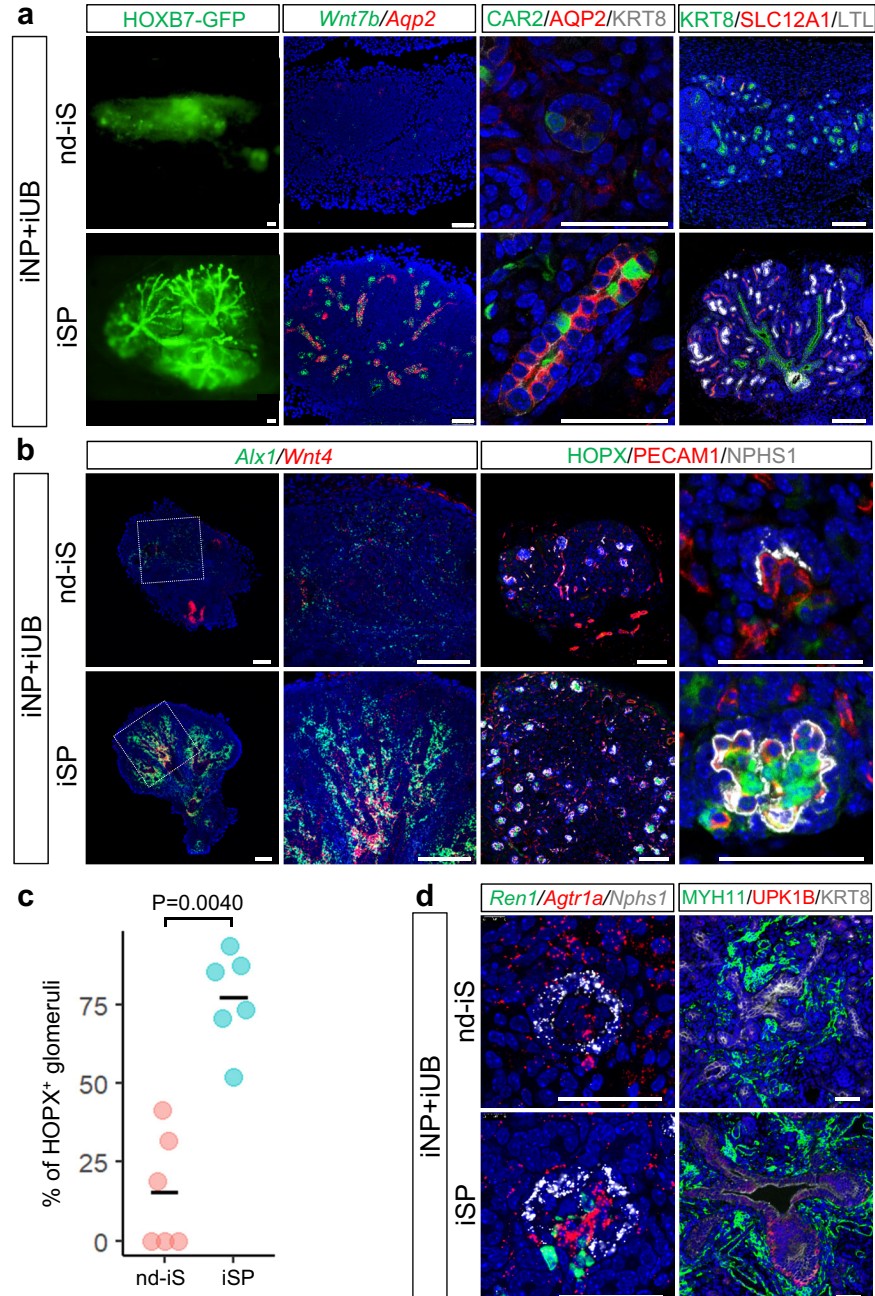

**Fig. 5 Multiple types of interstitial cells are differentiated in the ESC-derived kidney upon transplantation. a** Differentiation of UBs and nephrons in mouse ESC-derived transplanted organoids (generated using iSPs or nd-iS). 1st column: GFP images of UBs; 2nd column: in situ hybridization of *Wnt7b* (UB stalks) and *Aqp2* (principal cells); 3rd column: immunostaining of CAR2 (intercalated cells), AQP2 (principal cells), and KRT8 (UBs); 4th column: immunostaining of KRT8 (UBs), SLC12A1 (loops of Henle), and LTL (proximal tubules). Scale bars: columns 1–3, 50 μm; column 4, 100 μm. Six organoids in each condition obtained from three independent experiments were analyzed. **b** Differentiation of stromal cells. 1st column: in situ hybridization of *Alx1* (medullary stroma) and *Wnt4* (innermost medullar stroma); 2nd column: magnified images of the first columns; 3rd column: immunostaining of HOPX (mesangial cells), PECAM1 (ECs), and NPHS1 (glomerular podocytes); 4th column: magnified images of the 3rd columns. Scale bars: 100 μm. **c** Percentages of glomeruli equipped with HOPX$^+$ mesangial cells. Data are shown as mean ± SEM. Six organoids in each condition obtained from three independent transplantation experiments were analyzed. All organoids were serially sectioned, and two sections per organoid were stained to calculate the percentages of HOPX$^+$ mesangial cells. The non-parametric Mann–Whitney U test (two-sided) was performed. The source data are provided as a Source Data file. **d** Differentiation of mesangial cells and ureteric stroma. 1st column: in situ hybridization of *Ren1* (renin cells), *Agtra1* (mesangial cells), and *Nphs1* (podocytes); 2nd column: staining of MYH11 (ureteric stroma), UPK1B (ureter epithelium), and KRT8 (UBs). Scale bars: 50 μm. All of the data were obtained at day 14 post-transplantation, except for the GFP images of UBs obtained at day 10 post-transplantation. Six organoids in each condition obtained from three independent experiments were analyzed.

(Fig. 5b, 3rd and 4th columns). Inside these glomeruli, we detected cells positive for HOPX, a representative mesangial cell marker. The percentages of HOPX$^+$ glomeruli in organoids with iSPs were significantly higher than those in organoids with nd-iS (Fig. 5c). Angiotensin receptor type 1a (*Agtr1a*), another mesangial cell marker, was detected inside the glomeruli (Fig. 5d, 1st column), and *Ren1* (mouse *renin* gene) was detected at sites close to the glomeruli. In addition, *Ehd3*$^+$ glomerular capillaries, as well as *Gja5*$^+$ periglomerular arterioles and medullary arteries, were formed (Supplementary Fig. 4b), suggesting the establishment of functional vascularization in the organoids.

We also detected ureter-like structures, consisting of epithelia expressing UPK1B (urothelial marker) surrounded by stroma expressing MYH11 (ureteric stromal marker) (Fig. 5d, 2nd column). When harvested at a later timepoint (day 28 post-transplantation), marked dilatation of the ureter was observed (Supplementary Fig. 4c), suggesting possible urine accumulation in the ureter. Although we aimed to induce dorsal SPs, the expression levels of some intermediate and ventral SP genes remained higher than those in vivo (Fig. 3c and Supplementary Fig. 3c). This may have enabled iSPs to differentiate into the ureteric stroma at the UB stalk, thereby inducing the iUB epithelia to differentiate into UPK1B$^+$ urothelial cells. Thus, the ESC-derived iSPs differentiated into cortical and medullary stroma, intraglomerular mesangial cells, extraglomerular renin cells, and even smooth muscle cells around the ureter.

**Perinatal gene expression in the transplanted organoids.** We performed scRNA-seq analysis of the transplanted ESC-derived kidney (day 14 post-transplantation) and compared the data with those for embryonic kidneys during late gestation described in our previous report[36]. The analysis demonstrated the presence of multiple types of nephrons, UBs, and stroma in organoids generated using iSPs (Fig. 6a). The organoids using nd-iS also had complex clusters judging from the poor histological structures shown in Fig. 5, but the individual clusters were not as discrete as those in the iSP-organoids. In both types of organoids, we confirmed the ESC origin of all three renal lineages based on the sex difference between the donor ESCs (male) and the host animals (female). Female-specific *Xist* was undetectable in nephrons, UBs/collecting ducts, and stromal cells, but was detected in ECs, lymphocytes, and leukocytes (Fig. 6b). This is consistent with our previous finding that the host-derived vasculature integrates into organoids upon transplantation[35]. Apparent off-target clusters, such as neurons or muscles, were not detected (Fig. 6a), presumably because we purified the progenitors before aggregation.

The parenchymal clusters in iSP-derived organoids overlapped with the clusters in the E15.5 and neonatal (P0) kidney in vivo (Fig. 6a). The nephron clusters for glomerular podocytes, proximal and renal tubules, loops of Henle, and distal tubules, as well as the UB/collecting duct clusters of principal cells and intercalated cells, were clearly detected (Fig. 6a and Supplementary Fig. 5a). Maturation markers detectable at P0 but not E15.5[36], such as those for podocytes (*Col4a3*, *Htra1*), proximal tubules (*Slc5a2*, *Fbp1*), principal cells (*Aqp2*), and intercalated cells (*Atp6vg3*), were also expressed in the iSP-derived organoids (Supplementary Fig. 5a). In contrast, NPs (*Six2*$^+$) and UB tips (*Ret*$^+$) were rarely detected (Fig. 6a, arrows, Supplementary Fig. 5a), indicating depletion of the nephrogenic niche. This niche depletion was not specific to organoids, because it was also observed upon transplantation of embryonic kidneys[36], although the precise mechanism remains unclear. Glomerular ECs (*Ehd3*$^+$) and periglomerular arterioles (*Gja5*$^+$) were also detected, but the expression of some representative genes was lower than that in glomerular ECs in vivo (Supplementary Fig. 4d). Because the ECs were not derived from

mouse ESCs, these data imply the flexibility, as well as the limitation, of the adult ECs in the host animals.

Stromal cell clusters from the scRNA-seq analysis were extracted for further detailed analysis (Fig. 6c, d). The iSP-derived organoids showed mostly overlapping clusters with those in vivo (P0), including the cortical and medullary stroma, mesangial cells, renin cells, and smooth muscle cells around the ureter (Fig. 6c and Supplementary Fig. 5b). In contrast, in the organoids derived from nd-iS, only a small fraction of the cells contributed to the authentic cell types, and 43.9% of the cells (1152 cells in Clusters 0 and 9 out of 2626 cells in total) were positioned distantly from those in vivo (Fig. 6c). We also detected an ectopic cluster in the iSP-derived organoids (Fig. 6c), but with a smaller percentage (2.8%; 81 cells in Cluster 35 out of 2879 cells in total). An unbiased hierarchy clustering analysis also showed that the iSP-derived interstitial cells were similar to those in vivo than the nd-iS-derived interstitial cells (Fig. 6d and Supplementary Data 2). We selected representative markers for each stromal subpopulation in vivo (P0), and examined their expression levels in the organoids (Supplementary Data 1). UMAP and dot plot analysis showed that many stromal marker genes in the organoids were expressed at comparable levels to P0 (Fig. 6e and Supplementary Fig. 5b). Expression of SP-related genes was decreased, consistent with the nephrogenic niche depletion described above. Many stromal markers appeared to be expressed in the nd-iS-derived organoids but, as described above, only a small fraction of the cells contributed to the authentic cell types, and thus a small number of cells managed to differentiate despite the poor three-dimensional structure. The organoid-specific clusters expressed some ectopic genes, including *Pgr* and *Ngfr* (Fig. 6e, Supplementary Fig. 5b, and Supplementary Data 1). Taken together, ESC-derived kidney organoids generated by combining iNPs, iUBs, and iSPs, were organized with multiple types of stroma and parenchyma (nephrons and UBs/collecting ducts), and expressed many genes detected in neonatal kidneys.

## Discussion
By elucidating the molecular features of the in vivo renal stromal lineages at single-cell resolution, we established in vitro protocols to induce dorsal SPs from mouse ESCs. When ESC-derived SPs, NPs, and UBs were assembled, the organotypic kidney structure that consists of branching UBs with multiple nephrons was generated. Furthermore, the iSPs could differentiate into multiple types of stroma: cortical and medullary stromal cells, as well as mesangial cells and renin cells.

Our data revealed the importance of mediolateral and dorsoventral patterning in kidney specification. Our SP induction protocol involves SHH and a BMP inhibitor. Although the role of SHH in early embryogenesis was reported[28–30], the effect of SHH on the kidney lineage precursors, namely the IM, has remained unclear. SHH expressed in the midline medializes the mesoderm, while BMP in the LPM lateralizes the mesoderm. Because the IM is formed between the LPM and the PM (close to the midline), the balance between SHH and BMP is likely to be critical. Indeed, the IM expressed BMP antagonist *Grem1*, and manipulation of SHH and BMP signals enhanced SP induction in vitro. Our data further suggest that dorsoventral patterning is important for specifying the three types of SPs: FGF induced dorsal SPs, while BMP enhanced ventral gene expression. Indeed, these patterning concepts enabled the generation of dorsal SPs that exhibited similar gene expression profiles to their in vivo counterparts, as well as functionality, in sharp contrast to the non-parenchymal cells (nd-iS) in the conventional NP organoids.

When dorsal iSPs were combined with iNPs and iUBs, kidney tissue with the organotypic structure was obtained purely from

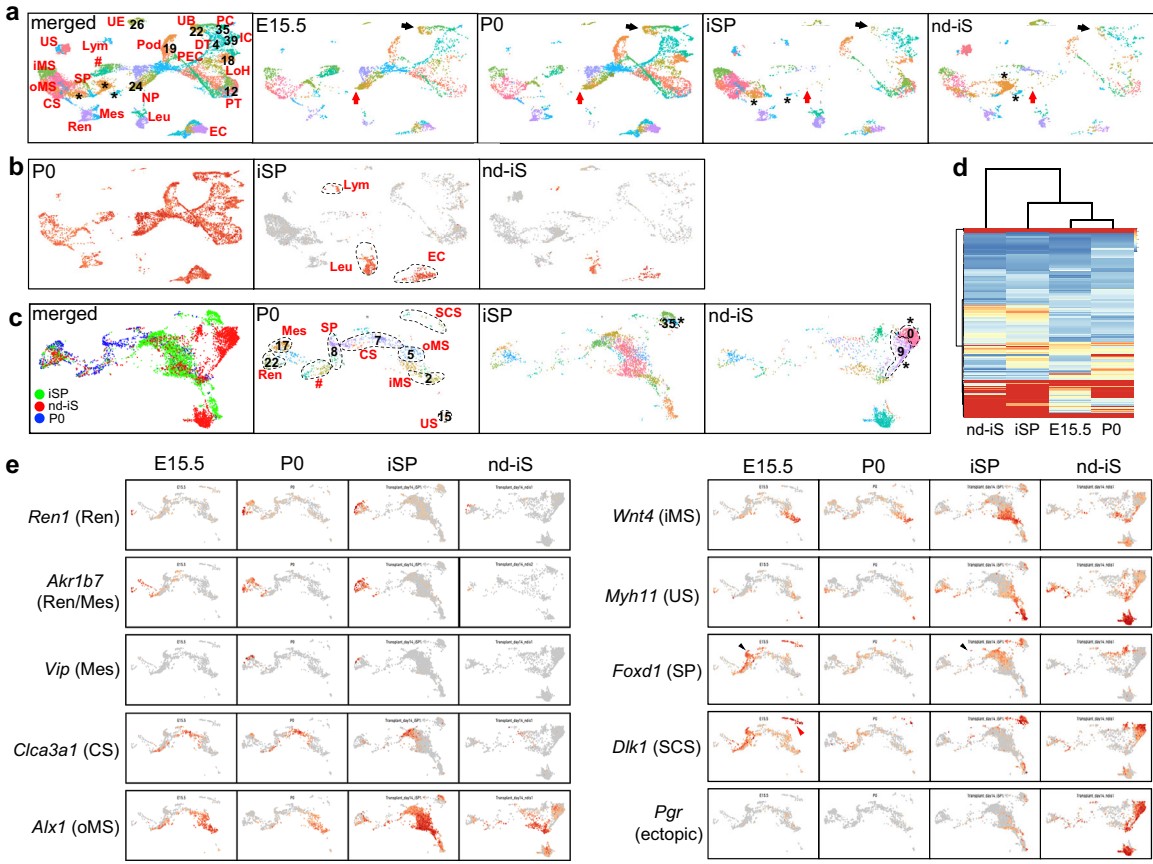

**Fig. 6 Nephron, UB, and stromal lineages in the ESC-derived kidney express perinatal renal genes. a** UMAP plots of embryonic kidneys (E15.5 and P0) and mouse ESC-derived transplanted organoids (generated using iSPs or nd-iS). The lack of NPs and UB tips in the transplanted organoids is indicated by red and black arrows, respectively. Pod: podocytes; PEC: parietal epithelial cells; PT: proximal tubule; LoH: loop of Henle; DT: distal tubule; PC: principal cells; IC: intercalated cells; UE: uroepithelium; Ren: renin cells; Mes: mesangial cells; CS: cortical stroma; oMS: outer medullary stroma; iMS: inner medullary stroma; US: ureteric stroma; Leu: leukocytes; Lym: lymphocytes; EC: endothelial cells; *: organoid-specific clusters; #: proliferating cells. **b** UMAP plots of *Xist*. *Xist* is absent in nephrons, UBs, and stroma derived from male mouse ESCs, but detected in the female host-derived lymphocytes (Lym), leukocytes (Leu), and ECs. **c** UMAP plots of extracted stromal cells in the P0 kidney and organoids. Mes: mesangial cells; Ren: renin cells; CS: cortical stroma; oMS: outer medullary stroma; iMS: inner medullary stroma; US: ureteric stroma; SCS: subcapsular stroma; *: organoid-specific clusters; #: proliferating cells. **d** Unbiased hierarchal clustering analysis of the induced stroma and embryonic stroma. **e** UMAP plots of representative genes in the stromal cells of the embryonic kidneys (E15.5 and P0), iSP-derived organoids, and nd-iS-derived organoids. Ren: renin cells; Mes: mesangial cells; CS: cortical stroma; oMS: outer medullary stroma; iMS: inner medullary stroma; US: ureteric stroma; SCS: subcapsular stroma (red arrowhead). Black arrowheads indicate the lack of SPs in the transplanted organoids.

mouse ESCs. This emphasizes the importance of stroma for the generation of the "higher-order structure" of the organ. As shown in our previous report[17], assembly of differentially induced progenitors is an effective approach to generate the organotypic structure, although a single induction protocol is frequently adopted in the organoid research field. Several groups recently reaggregated differentially induced human nephron organoids (theoretically containing NPs and nd-iS) and UB organoids and showed some mutual interactions[19–21]. However, properly branched UBs were not formed. Our mouse kidney organoids contained branching UBs in the center and multiple types of nephrons at the periphery, which was only achievable in the presence of authentic stroma. Although the precise interaction mechanisms between the stroma and parenchyma remain to be elucidated, previous loss-of-function studies in vivo have revealed some of the responsible genes/signals from stroma to parenchyma, including *Gdnf* and *Aldh1a2* for UBs, and *Fat4* for NPs[5,11,12]. Indeed, iSPs expressed many of these genes at high levels. Our building-up strategy using organoids will contribute to the identification of unknown molecules involved in the stroma-parenchymal interactions. ECs are not primarily required for the

initial formation of the higher-order structure, because ECs were excluded from the PDGFRA+ SP fraction before aggregation[17]. However, upon transplantation, ECs are likely to contribute to the maturation of stromal and parenchymal cells in glomeruli, and possibly to the corticomedullary patterning of the renal stroma in concert with UBs.

The present data demonstrate that iSPs can differentiate into multiple types of the stroma. The corticomedullary axis was formed in our organoids, and at least four layers of the stroma (*Fibin*+ outermost cortical stroma, *Lox*+ cortical stroma, *Alx1*+/*Wnt4*− medullary stroma, and *Alx1*+/*Wnt4*+ deeper medullary stroma) were generated. It was reported that *Wnt7b* in the UB stalks emits its signal to the surrounding medullary stroma and is required for renal medulla formation and nephron elongation in the medulla[37]. Our data suggest that such interactions between stroma and parenchyma occurred in the organoids. The formation of mesangial cells and renin cells was only observed upon transplantation, suggesting that the in vitro culture condition was not optimal for stromal differentiation. After transplantation, the glomeruli became vascularized and specialized stromal cells were formed, probably through interactions between SPs, ECs, and

glomerular podocytes. Renin is a secreted hormone produced by the juxtaglomerular apparatus that cleaves angiotensinogen in the plasma to generate angiotensin I[38]. Subsequent processing generates angiotensin II, which binds to its receptors, including AGTR1A expressed in mesangial cells. Mesangial cells contract in response to angiotensin II, thereby regulating glomerular filtration through glomerular capillaries[39]. Thus, the generation of renin cells and mesangial cells is a significant step toward functionality of the kidney. Recently, cAMP stimulation of conventional kidney organoids was shown to induce renin production[40]. Combination of our protocol with cAMP stimulation may further increase the induction efficiency for renin cells.

There remain several challenges toward functional kidney generation. Mouse kidney organoids lack an elongated ureter and remain unconnected to the host urinary outlet; thus, a complete urinary flow route is not established. Because we now know that the addition of BMP4 favors intermediate and ventral SP formation, it will become achievable to generate more elongated ureters with smooth muscle layers, which are descendants of *Tbx18*[+] intermediate SPs. It is also important to examine whether our SP induction protocol can be applicable to human iPSCs. The similarity in gene expression between mouse and human SPs remains unclear. For example, *FOXD1* was reported to be expressed in SPs and NPs at almost equal levels in humans[41]. Accumulating scRNA-seq data on human embryonic kidneys[41,42] will be valuable for the identification of SP markers in humans, and for precise assessment of the induced SPs and resultant stroma. For the higher-order kidney structure in humans, the quality of the induced NPs and UBs also needs to be fine-tuned based on information obtained in vivo.

Taken together, we have revealed the developmental program of renal SPs and successfully induced these cells in vitro. The inclusion of iSPs enabled the generation of organoids with the organotypic structure of the kidney equipped with multiple types of stromal cells. Our study provides a proof-of-concept for the generation of organoids with the "higher-order structure" derived purely from PSCs. It also serves as a useful platform to study the heterogeneous stromal cells in organs.

## Methods

**Mice**. *Osr1-GFP* mice and *Tbx18-MerCreMer* mice were generated as described previously[14,43]. *Hoxb7-GFP* mice[44], *Foxd1-GFPCre* mice[7], *Foxd1-GFPCreER^T2* mice[7], *Isl1-MerCreMer* mice[45], and *ROSA26-CAG-tdTomato* mice[46] were purchased from Jackson Laboratory. *Foxd1-GFPCreER^T2*, *Isl1-MerCreMer*, and *Tbx18-MerCreMer* mice were maintained on a C57BL/6 background, while the other strains were maintained on a mixed genetic background of C57BL/6 and ICR. For transplantation experiments, 8–10-week-old immunodeficient male mice (NOD/ShiJic-scidJcl) were purchased from Charles River Laboratory Japan Inc. The mice were housed in a specific pathogen-free animal facility in plastic cages on a 12 h/12 h light/dark cycle, and fed an irradiated CE-2 diet (CLEA Japan Inc.). All animal experiments were performed in accordance with institutional ethical guidelines and approved by the licensing committee of Kumamoto University (approval numbers: A2019–113 and A2021-008).

**Mouse ESC line maintenance**. Mouse ESC lines were maintained in Glasgow's minimal essential medium (Gibco #11710-35) supplemented with 14% knockout serum replacement (Gibco #10828028), 1% fetal bovine serum (FBS) (Japan Bio-serum #12303), 1% nonessential amino acids (Gibco #11140-050), 1% sodium pyruvate (Gibco #11360070), 0.1 mM 2-mercaptoethanol (Nakarai #21438-82), 1,000 U/ml leukemia inhibitory factor (Millipore #ESG1106), 1.5 μM CHIR, and 0.5 μM PD0325901. The mouse ESC lines were cultured in a humidified atmosphere containing 5% $CO_2$ at 37 °C. Cells were passaged every other day.

**SP induction from mouse ESCs**. The *Hoxb7-GFP* mouse ESC line (clone #B6-5; male) was described previously[17] and maintained on mitotically inactivated murine embryonic fibroblasts. The G4-2 mouse ESC line (male) with ubiquitous expression of GFP[47] was kindly provided by Dr. Hitoshi Niwa (Kumamoto University) and cultured on gelatin-coated plates. Dissociated mouse ESCs were aggregated and induced using the previously described NP induction protocol up to day 6.5[14,17]. Briefly, 1000 cells/50 μl were aggregated in 96-well U-bottom low cell-binding plates (Thermo Scientific #174925). After 48 h (at day 2), the spheres were

dissociated with Accutase (Millipore #SF006) and reaggregated in serum-free differentiation medium with human Activin A (1 ng/ml). After 24 h (at day 3), the medium was switched to medium containing CHIR (10 μM). After 36 h (at day 4.5), the medium was replaced with medium containing Y27632 (10 μM) and CHIR (10 μM). At day 5.5, the medium was changed to medium containing Activin A (10 ng/ml), BMP4 (3 ng/ml), CHIR (3 μM), RA (0.1 μM), and Y27632 (10 μM). At day 6.5, the spheres were dissociated with Accutase at 37 °C for 8 min and the ROBO2[+]PDGFRA[+] fraction was sorted. The sorted cells were resuspended in differentiation medium (20,000 cells/200 μl) in the presence of Y27632 (10 μM), RA (0.1 μM), FGF9 (10 ng/ml), SHH-N (1 μg/ml), and LDN193189 (25 nM), seeded into 96-well low-cell-binding U-bottom plates, and cultured until day 9.5. The composition of the differentiation medium was 75% IMDM (Gibco #21056-023) and 25% Ham's F12 medium (Gibco #11765-054) supplemented with 0.5 × B27 without RA (Gibco #12587-010), 0.5 × N2 (Gibco #17502-048), 0.05% BSA (Sigma #A4503), 0.5 mM ascorbic acid (Sigma #A4403), 1 × GlutaMAX (Gibco #35050-061), 0.5× penicillin/streptomycin (Gibco #15070-063), and $4.5 × 10^{-4}$ M 1-thioglycerol (Sigma #M6145).

**NP and UB induction from mouse ESCs**. NP and UB organoids were induced from mouse ESCs based on our published protocols[14,17], and were subjected to RNA-seq analyses. For NP organoids, the spheres at day 6.5 were cultured for 3 days in the presence of containing CHIR (1 μM), human FGF9 (5 ng/ml), and Y27632 (10 μM). For UB organoids, 1000 cells/50 μl were aggregated in 96-well U-bottom low cell-binding plates. After 48 h (at day 2), the spheres were dissociated with Accutase and reaggregated in differentiation medium with human Activin A (3 ng/ml). After 24 h (day 3), the medium was switched to medium containing human CHIR (10 μM). After 36 h (at day 4.5), the medium was changed to medium containing RA (0.1 μM), human Fgf9 (200 ng/ml), and CHIR (2 μM). At day 5.5, the medium was changed to medium containing RA (0.1 μM), human Fgf9 (100 ng/ml), and CHIR (3 μM). At day 6.25, induced spheres were dissociated by 0.25% trypsin/EDTA and sorted. A total of 3,000 sorted CXCR4[+]KIT[+] cells were aggregated in V-bottom 96-well low cell-binding plates (Sumitomo Bakelite #MS-9096V), and cultured in the presence of Y27632 (10 μM), RA (0.1 μM), CHIR (1 μM), human FGF9 (5 ng/ml), and 10% Matrigel (growth factor reduced; Corning #356230). At day 7.25, the medium was replaced with medium containing Y27632 (10 μM), RA (0.1 μM), CHIR (3 μM), human FGF9 (5 ng/ml), human GDNF (1 ng/ml), and 10% Matrigel. At Day 8.25, the spheres were transferred to medium containing Y27632 (10 μM), RA (0.1 μM), CHIR (3 μM), human GDNF (2 ng/ml), and 10% Matrigel, and cultured for 24 h (day 9.25). For the reconstitution assay, the last 3 steps (from day 6.25 to 9.25) of the UB protocol were modified to promote the formation of UB tips. After sorting of CXCR4[+]KIT[+] Wolffian duct progenitors, the cells were treated with Y27632 (10 μM), TTNPB (100 nM), Jak inhibitor I (10 μM), SB202190 (5 μM), CHIR (3 μM), A83-01 (200 nM), LDN193189 (200 nM), FGF9 (50 ng/ml), and GDNF (50 ng/ml) for 3 days, as described by Zeng et al[48].

**Kidney reconstitution from three types of progenitors**. NPs, UBs, and SPs were isolated from E11.5 mouse kidneys or induced from mouse ESCs. NPs and SPs were sorted based on the expression of ITGA8 and PDGFRA, and UBs were isolated as intact buds. To create a kidney organoid with the "higher-order structure", 50,000 SPs and 25,000 NPs were mixed and seeded into 96-well low-cell-binding U-bottom plates, followed by centrifugation (210 × $g$, 4 min). Single UBs isolated manually using tungsten needles were placed onto the deposited sheet-like cells. After culture at 37 °C overnight, the aggregated spheroids were transferred to Transwell inserts (Costar #3422) containing 50 μl of 50% Matrigel (growth factor reduced; Corning #356230) in DMEM/F12 (Gibco # 11320-033) with 10% FBS (Sigma #172012), and cultured for 7 days for in vitro analyses. The organoids were transplanted at day 3.

**Transplantation of the organoids**. The organoids were transplanted under the kidney capsule of the immunodeficient mice, as described[35,49]. Female NOD/SCID mice aged 8–10 weeks were used as the host animal and anesthetized with normal saline containing 0.75 mg/kg medetomidine, 4.0 mg/kg midazolam, and 5.0 mg/kg butorphanol. The host kidney capsule was incised and the 4% agarose (Sigma #6013) rods were inserted. Inserted rods were arranged to make a V-shaped free space and briefly cauterized with capsule membrane by electric cautery. One organoid was inserted from the incised window by a 24-gauge plastic indwelling needle connected by a P-200 pipette. After surgery, atipamezole was administered as an anesthetic antagonist.

**scRNA-seq analyses**. scRNA-seq data of E15.5 and P0 embryonic kidneys were described previously[36]. E9.5 embryos (caudal from the 26th somite), mouse embryonic kidneys (E11.5, E13.5), and the transplanted kidney organoids, were digested in 0.25% trypsin/EDTA for 10 min. Trypsin was inactivated by addition of DMEM (Sigma #D5796)/10% FBS containing 50 μg/ml DNase I (Worthington #LS002139), and the cells were washed with HEPES-buffered saline solution (HBSS) (Gibco #14185-052) containing 2% FBS, 50 μg/ml DNase I, 1 mM $CaCl_2$ (Wako #031-00435), and 0.035% $NaHCO_3$ (Wako # 191–01305). Cells were resuspended in 0.04% bovine serum albumin (BSA)/phosphate buffered-saline

(PBS), filtered through a 40 μm-pore strainer (Falcon; Cat# 352340), and evaluated for their cell number and viability (>90%) using a Countess automated cell counter (Invitrogen # C10227).

Aliquots containing 5000 dissociated cells from each sample were applied to a Chromium Controller (10× Genomics). A Chromium Single Cell 3′ Library & Gel Beads Kit v2 or v3 (10× Genomics) was used to generate cDNA libraries, which were then sequenced by an Illumina HiSeq 3000 (446,005,466 reads for mouse E9.5; 672,154,533 reads for E11.5; 311,733,555 reads for E13.5; 757,548,299 reads for E15.5; 955,496,397 reads for P0; 434,091,903 reads for mouse iSPs; 419,265,793 reads for mouse nd-iS; 387,644,117 reads for mouse iNPs; and 404,051,455 reads for mouse iUBs. The Q30 base RNA reads (Q-scores indicating sequencing quality) of the samples were 90.5% for E9.5, 82.9% for E11.5, 88.5% for E13.5, 86.2% for E15.5, 93.6% for P0, 92.1% for iSPs, 91.6% for nd-iS, 85.2% for iNPs, and 85.9% for iUBs.

The raw sequence data were processed using the *cell ranger count* command in Cell Ranger (version 2.0.0 for E9.5, 3.0.2 for E11.5, and 4.0.0 for integrated data; 10× Genomics). Subsequent analyses were performed using the R statistical programming language[50] and the Seurat package (version 3.1.1 for E9.5 and E11.5, and version 4.0.0 for integrated data)[51,52]. The final dataset contained 15,434 genes and 3794 cells for E9.5, 14,885 genes and 1294 cells for E11.5, 28,410 genes, and 30,760 cells for integrated mouse data. A principal component analysis was applied for dimension reduction with dimension values of 46 for E9.5 and E11.5, 95 for integrated mouse data, determined by the JackStrawPlot function[53]. Uniform Manifold Approximation and Projection for Dimension Reduction (UMAP) plots were generated using the uwot package[54].

**Gene extraction from UMAP clusters and dot plot analysis**. We generated representative genes in each cluster using the FindMarkers function in Seurat[52]. We compared each target cluster with all other clusters at P0 and selected the genes with low *P*-values (*P* < 0.05) and low expression rates in other clusters (pct2 < 0.25). We then verified the expression of these genes in UMAP plots. Dot plot analysis was performed using the DotPlot function in Seurat.

**Unbiased hierarchal clustering analysis**. We initially obtained stromal clusters using the subset function, and then calculated the average expression of all genes detected at each stage (E15.5, P0, iSPs, and nd-iS) using the AverageExpression function in Seurat[52]. Subsequently, representative genes for the P0 stroma were selected (Supplementary Data 2) by the method described above. The pheatmap function[55] was used to obtain a heatmap with hierarchy.

**Flow cytometric analysis**. Organoids induced from embryonic tissues or mouse ESCs were dissociated and blocked with normal mouse serum for 10 min on ice. Cell surface marker staining was carried out in 1× HBSS containing 1% BSA and 0.035% NaHCO₃ for 15 min on ice. Stained cells were analyzed using a FACS CANTO II (BD Biosciences) or a FACS SORP Aria (BD Biosciences). The original FACS data sets were collected by using BD FACS Diva software (v8.0.1). Data analyses were performed with FlowJo software (ver 7.6.5, TreeStar). Quantification data were presented as mean ± SEM. Detailed antibody information is provided in Supplementary Table 1. The FACS sequential gating/sorting strategies are shown in Supplementary Fig. 6.

**Isolation of the posterior IM in embryos**. We sorted the Osr1-GFP⁺ fraction or ROBO2⁺/PDGFRA⁺ fraction after removing the hindlimb buds and LPM from E9.5 mouse embryos (caudal from the 26th somite; somite at the caudal end of the forelimb defined as somite 15).

**Immunohistochemistry**. For whole-mount staining, organoids were fixed in 4% paraformaldehyde (Sigma #158127) for 60 min, washed with 0.1% Triton X-100 (Nakarai #35501-15) in PBS three times, and blocked in PBS containing 10% goat serum (Nippon Bio-Test Laboratories), 1% Triton X-100, and 2% dry skim milk (Cell Signaling #9999 S) for 1 h twice, as described previously[17]. The samples were then incubated with primary antibodies at 4 °C overnight, washed with 1% TritonX-100 for 1 h three times, and incubated with secondary antibodies conjugated with Alexa Fluor dyes at room temperature for 2 h. After immunostaining, the tissues were clarified using ethyl cinnamate (Sigma #C12372)[56]. Three-dimensional fluorescence images were captured by confocal microscopy (TSC SP8; Leica) and reconstructed by Imaris (v7.7.0, Bitplane). For section immunostaining, paraffin sections were subjected to antigen retrieval in a citrate buffer, washed three times with PBS, and blocked by incubation with 1% BSA in PBS for 1 h at room temperature. The sections were incubated with primary antibodies at 4 °C overnight, followed by incubation with secondary antibodies conjugated with Alexa Fluor dyes at room temperature for 2 h. Fluorescence images were captured by confocal microscopy (TSC SP8; Leica). Detailed antibody information is provided in Supplementary Table 1.

**Section in situ hybridization**. RNAscope analysis of 10% formalin-fixed paraffin sections was performed using an RNAscope Multiplex Fluorescent Reagent Kit v2 (Advanced Cell Diagnostics; Cat# 323100). Signal amplification was performed

with TSA plus fluorophores (Thermo Fisher Scientific). Details of the RNAscope probes are provided in Supplementary Table 1.

**RNA extraction and quantitative RT-PCR**. Total RNA was isolated using an RNeasy Plus Micro Kit (Qiagen #74034), and reverse-transcribed using Superscript VILO cDNA synthesis kit (Invitrogen #11754-250). Quantitative PCR was carried out using a Real-Time PCR System (Takara Bio) and TB Green Fast qPCR Mix (Takara Bio #RR430A). Relative mRNA expression levels were normalized to β-actin gene expression. Data below the detection limits were plotted on the baseline of the bar charts. Detailed primer information is provided in Supplementary Table 2.

**Quantification and statistical analysis**. Quantification data, including qRT-PCR, were presented as mean ± SEM with plots. Bar charts with plots were generated by using the R statistical programming language[50], ggpubr[57], and ggbeeswarm[58]. Two-sided Student's *t*-test or the non-parametric Mann–Whitney U test (two-sided) was applied for statistical analysis of differences between two groups. Differences with values of *P* < 0.05 were considered statistically significant. For statistical analysis of more than two groups, one-way ANOVA was performed. If the ANOVA was significant, multiple comparisons between groups were performed by Dunnett's multiple comparison test (two-sided) or the Tukey–Kramer test (two-sided) using Excel software.

**Reporting summary**. Further information on research design is available in the Nature Research Reporting Summary linked to this article.

## Data availability
All data supporting the conclusions are present in the paper and the supplementary materials. A reporting summary for this article is available as a Supplementary Information file. Figures 1, 2, 3, and 6 have associated scRNA-seq raw data, which have been deposited in the National Center for Biotechnology Information Gene Expression Omnibus (GSE178263). The scRNA-seq data of E15.5 and P0 embryonic kidneys were described previously (GSE149134)[36]. Source data are provided with this paper.

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

## Acknowledgements
We thank S. Fujimura, S. Usuki, M. Yamane, T. Ichikawa, K. Yasunaga, H. Naganuma, and S. Kuraoka for experimental assistance. We also thank Alison Sherwin, PhD, from Edanz Group (https://jp.edanz.com/ac) for editing a draft of this manuscript. This work was supported in part by a KAKENHI Grant (JP21H05050) from the Japan Society for the Promotion of Science and the Research Center Network for Realization of Regenerative Medicine (21bm0804013h0005) from the Japan Agency for Medical Research and Development.

## Author contributions
Conceptualization, R.N. and A.T.; methodology, S.T., E.T., and A.T.; investigation, S.T., E.T., K.M., T.O., D.I., and A.K.; resources, R.N. and CL.C.; writing—original draft, S.T., E.T., and R.N.; writing—review & editing, A.K., A.T., and R.N.; funding acquisition, R.N.; project administration and supervision, R.N.

## Competing interests
The authors declare no competing interests.
