## [Peer Review File · Nature Communications]

Generation of the organotypic kidney structure by integrating pluripotent stem cell-derived renal stromaREVIEWER COMMENTS

Reviewer #1 (Remarks to the Author):

In this manuscript, Tanigawa and colleagues have generated a refined kidney organoid protocol that includes molecularly correct stromal progenitors combined with nephron progenitors and ureteric bud progenitors. The result is a structure that more closely resembles a kidney both in terms of anatomy and parenchymal maturation than any of the previously published protocols. I find the data to be convincing and of high quality and the conclusions to be quite remarkable. This protocol will certainly become the standard in the field. My only wish is that they had applied this protocol to human iPSC or ES cell derived organoids. Hopefully that data is coming.

My only issues with this manuscript are minor. Some of the figures appear to be missing p-values (figure 2j and supp fig 2i) and some terms in the figures are not described in the figure legends. The authors should go over their figures/legends in detail prior to resubmission. These issues should be easy to fix. Beyond that, I find this to be outstanding work and strongly encourage publication at Nature Communications.

Reviewer #2 (Remarks to the Author):

The manuscript present by Tanigawa and Tanaka et al reported a new protocol to generate kidney organoid by incorporating the induced stromal progenitors (iSPs), the induced nephron progenitors (iNPs) and the induced ureteric buds (iUB) – all are generated from pluripotent stem cells (PSC) in vitro. This is an extended study of the previous high-order kidney organoid protocol reported by the same team (Taguchi et al, Cell Stem Cell 2017), where in the present manuscript the authors replaced the embryonic SPs with iSPs. There are three key contributions to the kidney development/regeneration field based on the data showing in this manuscript: First, the authors delineated the heterogeneity of embryonic SPs by performing single cell RNA-seq on a E11.5 kidney and validated that the FGF, BMP and RA signals are critical for dorsoventral patterning of the stroma in the embryonic kidney. Second, the author developed two protocols to induce dorsal SPs from embryonic posterior IM and embryonic stem cells. Third, the author developed and validated a new kidney organoid protocol that can generate organoids with high-order kidney structure. The results present by the authors are overall very convincing due to the fact that all experiments are very well-designed and the authors employed multiple approaches to carefully validate the data such as the use of in vivo, ex vivo and in vitro experiments to support the role of dorsal SPs in UB branching and the use of scRNA-seq, IF/ISH and reporter mice/cell lines to monitor and trace the cell populations in each tissue/organoid generated by the new protocols. In short, this is an outstanding study of its kind and will potentially receive a lot of interests from the peer researchers. The main issue of the current study, however, is that the data from the scRNA-seq analysis are not of high quality. Therefore, my comments below mainly highlight the problems in the scRNA-seq analysis part, and hopefully these comments can help the authors improve the quality of the current manuscript.

1. For all UMAP plots present in this manuscript, the authors only give cell names to those clusters that they think are critical to support their conclusions. This is an biased way to present the data from the scRNA-seq technique, which itself is an unbiased approach. There are a number of caveats which concern the validity of the advantages of the iSPs protocol over other prevailing protocols in the field. For example, previous studies have indicated that a substantial number of non-kidney cell types (primarily neurons and muscle cells) are also generated in the kidney organoid protocols (PMID: 30573816, PMID: 30449713, and PMID: 31784515), whether the

unannotated cell clusters in the present organoid protocol are off-target cells? Whether those unknown clusters are immature cell types, or just some low quality cells? The authors are strongly encouraged to define all cell types in their single cell datasets.

2. The comment above brings up another concern: have the authors examined the limitations of the iSP+iNP+iUB protocol? For example, what percentage of cells are off-target cells (non-kidney cell types) and what percentage of cells are immature cell types (e.g. expressing the progenitor cell markers or cell cycling genes)? Can another important glomerular cell type— endothelial cells— be also differentiated from the current protocol?

3. From Figure 1b-d, the authors described the heterogeneity of the embryonic SP based on the expression of several genes in different location of the clusters. A proper way to do this is to pull out the cells from the embryonic SP and re-cluster them using the unsupervised methods (e.g. Seurat). This transcriptome-wise classification of subclusters is more accurate than manual inspection of several marker genes.

4. The authors concluded that iSPs are more similar to the native dorsal SPs than the nd-iS (page 8, line 239-240) based on the cell overlapping from the UMAP 2D space (figure 3d). Since the cell-cell distance in UMAP is altered by the parameters chosen (for example, the min_dist and the metric parameters), using the visual inspection of overlapping effect from UMAP to assess the cell type similarity is inaccurate. A proper way to do this is to compute the pearson correlation and use the coefficient (pearson's r) value to assess the similarity. The authors can also consider using other more complicated algorithms to do this, such as scMAP (PMID: 29608555), scID (PMID: 32151972) or methods using the Random Forest Classifier.

5. In Figure 6a, although I understand that the figure was generated from integrated analysis of 4 datasets and then was split based on the origins of cells, it will be more informative if the authors can show another UMAP graph with all cells included and all cell types annotated because the current figure does not tell us how the clustering looks like and how many cell types are resolved.

6. In Figure 6e, the authors only compared the gene expression between iSPs and P0 kidney. It is important to include the nd-iS group and show that the cells from nd-iS group do not express or express lowly those genes.

7. Since the study would be of much higher value if the current dorsal SP induction protocol from mouse can be applied to human, the authors should discuss more about what would expect if researchers want to develop a similar protocol for human kidney organoid based on the current protocol.

Reviewer #3 (Remarks to the Author):

This is a beautiful paper where the authors advance the field by dissecting stromal cell differential in the developing kidney in the mouse and subsequently applying this insight into a in vitro differentiation protocol of kidney organoids that results in better 3D structural organization. The experiments are technically well performed and use Tate of the art techniques.

In fact, I have only major comment, but one that needs to be addressed. The authors do not mention the vasculature in this higher order kidney organoid at all. When transplanting the organoids under the kidney capsule, further maturation of stromal cells into perivascular phenotypes is described. I assume that functional vascularisation may have occurred and there is also literature that it is in fact this vascular patterning that drives the 3D structure of organs. So, can you describe what happens with the vasculature in your experiments and do you think that

that would be of relevance for the stromal patterning ?

If there were functional vascularisation (and hence filtration), where did the filtrate go. Did the authors observe cyste formation, as has been described with transplantation of embryonic mouse anlagen ? Particularly so, as the authors claim the organization into ureteric tissue.

Finally, it should be made clear from the title and abstract that this paper is about mouse embryonic development and mouse ES lines, and it still needs to be researched whether this insight also applies to human iPSC.

Generation of the organotypic kidney structure by integrating pluripotent stem cell-derived renal stroma

Shunsuke Tanigawa^{1#}, Etsuko Tanaka^{1#}, Koichiro Miike¹, Tomoko Ohmori¹, Daisuke Inoue¹, Chen-Leng Cai², Atsuhiko Taguchi^{1,3}, Akio Kobayashi¹, and Ryuichi Nishinakamura^{1*}

¹Department of Kidney Development, Institute of Molecular Embryology and Genetics, Kumamoto University, Kumamoto 860-0811, Japan

² Department of Pediatrics, Indiana University School of Medicine, Indianapolis, IN 46202, USA

³Present address: Department of Genome Regulation, Max Planck Institute for Molecular Genetics, Berlin, Germany

#These authors contributed equally.

*Corresponding author: Ryuichi Nishinakamura MD, PhD, Department of Kidney Development, Institute of Molecular Embryology and Genetics, Kumamoto University, 2-2-1 Honjo, Chuo-ku, Kumamoto 860-0811, Japan.

E-mail: ryuichi@kumamoto-u.ac.jp

Fax: +81-96-373-6618; Tel: +81-96-373-6615

Summary

Organs consist of the parenchyma and stroma, the latter of which coordinates the generation of organotypic structures. Despite recent advances in organoid technology, induction of organ-specific stroma and recapitulation of complex organ configurations from pluripotent stem cells (PSCs) have remained challenging. By elucidating the *in vivo* molecular features of the renal stromal lineage at a single-cell resolution level, we herein established an *in vitro* induction protocol for stromal progenitors (SPs) from **mouse** PSCs. When the induced SPs were assembled with two differentially induced parenchymal progenitors (nephron progenitors and ureteric buds), the completely PSC-derived organoids reproduced the complex kidney structure, with multiple types of stromal cells distributed along differentiating nephrons and branching ureteric buds. Thus, integration of PSC-derived lineage-specific stroma into parenchymal organoids will pave the way toward recapitulation of the organotypic architecture and functions.

Keywords

Organoid, kidney, stroma, parenchyma, stromal progenitor, pluripotent stem cell, ureteric bud, nephron progenitor, higher-order structure

Introduction

Recent progress in stem cell biology has enabled the generation of “organoids” from pluripotent stem cells (PSCs), but most of the currently available organoids lack the organotypic “higher-order structure”, in which functional units are distributed at the periphery of a branching epithelial backbone. One of the essential pieces for this organization is the stroma, which develops with epithelial parenchymal cells. However, previous studies mainly focused on generating the parenchyma and paid little attention to the authenticity of the stroma co-induced in the organoids¹. Because it has become apparent that the stroma of individual organs exhibits distinct characteristics and gene expression patterns²⁻⁴, we focused on the development of organ-specific stroma.

The embryonic kidney, the metanephros, forms its complex structure by mutual interactions of the metanephric mesenchyme (MM; including nephron progenitors [NPs] and stromal progenitors [SPs]) and the ureteric bud (UB)⁵⁻⁸. NPs and UBs are the parenchymal precursors. NPs form the capping mesenchyme around the UB tips and secrete GDNF to induce continuous branching of UBs that eventually develop into the urine-collecting system: the collecting ducts and ureter. In turn, UBs emit WNT and FGF signals that maintain NPs while inducing a subset of NPs to differentiate into nephrons composed of glomeruli and renal tubules^{9,10}. Meanwhile, SPs located in the periphery of the metanephros produce GDNF and ALDH1A2, the latter of which maintains GDNF receptor Ret expression in UB tips via retinoic acid (RA) signaling to support UB branching¹¹. SPs also express FAT4 to suppress excessive NP proliferation¹², allowing balanced nephron differentiation. In addition to their roles in UB branching and NP differentiation, SPs themselves differentiate into multiple types of interstitial cells. Lineage-tracing experiments showed that SPs expressing the transcription factor Foxd1 differentiate into the majority of the intrarenal stroma (interstitial cells), which is organized in a corticomedullary manner and surrounds the NP- and UB-derived epithelial parenchyma^{3,7}. SPs also differentiate into functionally specialized stromal cells such as mesangial cells and renin cells, which regulate glomerular filtration and systemic blood pressure, respectively^{7,13}. Thus, this triad (NP-UB-SP) interaction is essential for formation of the organotypic “higher-order structure” of the kidney: numerous nephrons distributed at the periphery of branching UB/collecting ducts, all surrounded by multiple types of stromal cells.

We and others reported the induction of kidney organoids from **mouse and human** PSCs¹⁴⁻¹⁶. Most of these studies involved induction of NPs, resulting in nephron formation in the organoids (NP organoids). We subsequently reported a protocol for selective induction of UBs, which show branching morphogenesis in gels (UB organoids), from mouse embryonic stem cells (ESCs) and human induced pluripotent stem cells (iPSCs)¹⁷. Notably, when NPs and UBs differentially induced from mouse ESCs were combined with stromal cells isolated from mouse embryonic

kidneys, the organotypic “higher-order structure” was reproduced, namely the peripheral progenitor niche and internally differentiated nephrons interconnected by a dichotomously branching ureteric epithelium. However, this structure was not formed in the absence of stromal cells¹⁷, as expected from the accumulated knowledge *in vivo*^{12,18}. Recently, several groups generated human iPSC-derived UB organoids and combined them with NP organoids¹⁹⁻²¹, but proper UB branching or differentiation of kidney-specific stromal cell types was not observed. Although all conventional organoids contain non-parenchymal “stroma-like” cells, their identity and differences from the *in vivo* counterparts remain obscure. Thus, the establishment of protocols for SP induction from PSCs precisely following the *in vivo* developmental trajectory is essential to recapitulate the generation of kidney-specific stromal cells and the organotypic architecture.

We previously identified spatiotemporally distinct origins for the MM (NPs+SPs) and the UB in mice¹⁴, which enabled us to establish induction protocols for NP and UB organoids^{14,17}. The T^+ immature mesoderm at embryonic day (E) 7.5 turns into the $Osr1^+$ anterior intermediate mesoderm (IM) at E8.5, and eventually differentiates into the UB. Meanwhile, the MM precursor is maintained in the caudal T^+ immature state for a longer period up to E8.5, and then differentiates into the $Osr1^+$ posterior IM at E9.5. As the $Osr1^+$ posterior IM further develops into $Six2^+$ NPs and $Foxd1^+$ SPs at E11.5^{14,22}, we hypothesized that modification of our NP induction protocol after the E9.5 posterior IM stage would lead to establishment of SP induction and eventually generate the “higher-order structure” of the kidney. To achieve this goal, we used the reverse induction approach established in our previous NP and UB lineage induction studies^{14,17}. Briefly, we initially focused on the last step of SP differentiation (from E9.5 $Osr1^+$ posterior IM to E11.5 $Foxd1^+$ SPs) and then combined the protocol with that for the earlier stages (from mouse ESCs to posterior IM). Finally, the obtained ESC-derived SPs were combined with ESC-derived NPs and UBs to examine the generation of the organotypic kidney structure, as well as the differentiation into kidney-specific stromal cell types.

Results

Dorsal SPs are essential for UB branching and intrarenal stroma formation

In a previous study¹⁷, we showed that combining E11.5 embryonic kidney-derived PDGFRA⁺ stromal cells with NPs and an isolated UB reconstituted the well-branched collecting duct structure *in vitro*. Because the *Foxd1*⁺ subpopulation among PDGFRA⁺ stromal cells gives rise to most of the kidney stroma⁷, we sorted GFP⁺PDGFRA⁺ SPs from E11.5 *Foxd1-GFP* mice and aggregated them with NPs and UBs (Supplementary Fig. S1a, b). The results showed that *Foxd1*⁺ SPs induced UB branching more efficiently than *Foxd1*⁻ cells (Supplementary Fig. S1c), proving that inclusion of *Foxd1*⁺ SPs was essential for generation of the organotypic structure. Immunostaining of the E11.5 kidney revealed that stromal domains expressing FOXD1, TBX18, and ISL1 were aligned along the dorsoventral axis, with the FOXD1⁺ domain located most dorsally and adjacent to SIX2⁺ NPs (Fig. 1a). We genetically labeled the three stromal domains at this stage using *Foxd1CreER*^{T2}, *Tbx18MerCreMer*, and *Isl1MerCreMer* mice, and confirmed that *Foxd1*⁺ SPs contributed to the majority of the intrarenal stroma at E15.5, *Tbx18*⁺ SPs mainly contributed to the stroma surrounding the ureter, and *Isl1*⁺ SPs contributed to the lower urinary tract including the bladder and urethra (Supplementary Fig. S1d), consistent with previous reports^{7,23,24}.

FGF and BMP signaling regulates dorsoventral patterning of renal SPs

To understand the development of SPs along the dorsoventral axis, we performed single-cell RNA-sequencing (scRNA-seq) of the E11.5 kidney, which revealed separate clusters of NPs (*Six2*⁺), UBs (*Ret*⁺), stromal cells (*Pdgfra*⁺), and endothelial cells (ECs; *Pecam1*⁺) in UMAP plots (Supplementary Fig. S1e). **Re-clustering of the extracted stromal cells showed that the stromal cells could be further divided into five subdomains: two with proliferation-related signatures and three likely representing *Foxd1*⁺, *Tbx18*⁺, and *Isl1*⁺ progenitors** (Fig. 1b, c). Marker genes for the three populations were also identified (Fig. 1c, d). Several signaling-related genes were expressed in the stroma: *Rarb* (RA signal indicator) in the entire stroma, *Etv4* and *Etv5* (FGF signal indicators) in the *Foxd1*⁺ domain, and *Id1* and *Smad7* (BMP signal indicators) in the *Tbx18*⁺ and *Isl1*⁺ domains (Fig. 1e). The expression of signaling ligands/synthesis enzymes (Fig. 1e, Supplementary Fig. S1e) was consistent with previous *in vivo* studies, including *FGF20* expression in NPs, *FGF9* in NPs and UB tips¹⁰, *BMP4* in the ventral stroma adjacent to the Wolffian duct²⁵, and *Aldh1a2* in the cortical stroma¹¹. Thus, we hypothesized that dorsoventral signal gradients may exist in the developing renal stroma: higher FGF signal dorsally, higher BMP signal ventrally, and ubiquitous RA signal throughout the stroma.

A previous cell lineage study demonstrated that NPs and SPs are derived from *Osr1*⁺ cells,

present in the IM and part of the lateral plate mesoderm (LPM) at E9.5²². To examine the FGF- and BMP-driven dorsoventral patterning of the stroma, we surgically removed the LPM and hindlimb buds from *Osr1-GFP* mice at E9.5, sorted the GFP⁺ posterior IM cells, and cultured them as spheres for 2 days in the presence of the above-mentioned factors as well as Y27632 to support cell survival (Fig. 1f). Treatment of GFP⁺ posterior IM cells with FGF9 and a Wnt agonist CHIR99012 (CHIR) was previously shown to result in NP generation, while addition of RA inhibited the NP induction¹⁴. In the present study, RA treatment of the same population enhanced two dorsal SP markers, *Foxd1* and *Hic1*, in addition to general stromal markers *Pbx1* and *Fat4* (Fig. 1g, Supplementary Fig. S1f). Addition of FGF9 to Y27632 and RA (YRF) further increased dorsal SP markers (*Foxd1*, *Crabp1*, *Ebf1*, *Asb4*), while ventral markers (*Hoxa13*, *Gata2*) were mildly decreased (Fig. 1h, Supplementary Fig. S1g). In contrast, BMP4 addition (YRB) markedly reduced dorsal SP markers (*Foxd1*, *Crabp1*), while markers for intermediate SPs (*Tbx18*, *Creb5*) and ventral SPs (*Isl1*, *Gata2*) were upregulated (Fig. 1i Supplementary Fig. S1h). Thus, two signaling pathways are likely to regulate the dorsoventral patterning of renal SPs: FGF9 induce dorsal SPs, while BMP4 induces ventral and intermediate SPs.

Sorting based on ROBO2/PDGFR α expression enables isolation of the posterior IM

To further evaluate the differentiation efficiency from the posterior IM toward the dorsoventral SPs, we aimed to sort the posterior IM from *Foxd1-GFP* mouse embryos. Because *Foxd1-GFP* is not expressed in the posterior IM population at E9.5, we searched for cell surface markers of the posterior IM. An scRNA-seq analysis of the posterior part (caudal from the 26th somite) of E9.5 embryos showed that the T⁺ tail bud mesenchyme segregated into neurons and somites (paraxial mesoderm [PM]). The LPM bifurcated into splanchnic (visceral) mesoderm and somatic (parietal) mesoderm (Fig. 2a, Supplementary Fig. S2a). *Osr1* was strongly expressed in the two clusters (IM and LPM), and weakly detectable in some dispersed cells adjacent to the presomitic mesoderm (PSM) (Fig. 2b). *Wt1* showed similar, but more restricted, expression patterns. *Grem1*, known to be expressed in the IM and MM^{26,27}, was detected in the IM cluster, but not in the LPM. *Six2* expression was more restricted to a portion of the IM cluster. We noted that expression of transmembrane receptor *Robo2* mostly overlapped with the *Osr1*⁺ IM and LPM domains (Fig. 2b). *In situ* hybridization on a posterior region similar to that used for the scRNA-seq analysis confirmed that *Osr1* and *Robo2* mostly overlapped in the IM and LPM (Fig. 2c, Supplementary Fig. S2b). *Grem1* was specifically expressed in the posterior IM that co-expressed *Osr1* and *Robo2* (Fig. 2c, Supplementary Fig. S2b).

UMAP plots showed that *Robo2*-positive neural tubes were negative for *Pdgfra*, while *Pdgfra* was expressed in the posterior IM, LPM, and PSM (Fig. 2b). Thus, we decided to test the combination of ROBO2 and PDGFRA for sorting the posterior IM. Because *Osr1* and *Robo2*

were also expressed in the LPM, we surgically removed the LPM and hindlimb buds from the posterior part of E9.5 *Osr1-GFP* mice, and analyzed ROBO2/PDGFR α expression by flow cytometry (Fig. 2d). The ROBO2⁺PDGFR α ⁺ fraction largely consisted of *Osr1-GFP*^{high} cells, while the ROBO2⁻PDGFR α ⁺ fraction exhibited varying degrees of *Osr1-GFP* expression (Fig. 2e, Supplementary Fig. S2c). The expression levels of IM marker genes (*Osr1*, *Grem1*, *Wt1*) were high in the ROBO2⁺PDGFR α ⁺ fraction (Fig. 2f), while PSM-related genes were enriched in the ROBO2⁻PDGFR α ⁺ fraction (Supplementary Fig. S2d). Thus, the method allowed successful isolation of the posterior IM without genetic fluorescent labeling.

Induction of dorsal SPs from the embryonic posterior IM

We sorted the ROBO2⁺PDGFR α ⁺ posterior IM and ROBO2⁻PDGFR α ⁺ non-IM fractions from wild-type mice and cultured them in the presence of RA and FGF9 (YRF), based on the results shown in Fig. 1. After 2 days of culture, the sizes of the spheres from the ROBO2⁺PDGFR α ⁺ fraction and *Osr1-GFP*⁺ posterior IM fraction were significantly larger than those from the ROBO2⁻PDGFR α ⁺ fraction (Fig. 2g, upper panels and right graph). A discrete PDGFR α ⁺ putative SP population was generated from the ROBO2⁺PDGFR α ⁺ fraction, as well as from the *Osr1-GFP*⁺ posterior IM fraction, but not from the ROBO2⁻PDGFR α ⁺ non-IM fraction (Fig. 2g, lower panels). We then examined the gene expressions in the induced PDGFR α ⁺ putative SPs. The ROBO2⁺PDGFR α ⁺ IM-derived cells showed higher expression levels of pan-stromal genes and dorsal SP genes, as well as intermediate and ventral SP genes, than the ROBO2⁻PDGFR α ⁺ non-IM-derived cells (Fig. 2h, Fig. S2e), and their expression levels were almost comparable to those in cultured *Osr1-GFP*⁺ IM cells and freshly isolated E11.5 dorsal SPs *in vivo*. Furthermore, when the PDGFR α ⁺ putative SPs derived from the ROBO2⁺PDGFR α ⁺ IM fraction were combined with E11.5 NPs and UBs, robust UB branching was observed (Fig. 2i, Supplementary Fig. S2f). Similar experiments were not achievable from the ROBO2⁻PDGFR α ⁺ fraction because few PDGFR α ⁺ cells were obtained after culture. Therefore, **the ROBO2⁺PDGFR α ⁺ posterior IM can be induced to renal dorsal SP-like cells *in vitro*.**

We examined the actual proportion of *Foxd1*⁺ cells induced in our culture conditions by utilizing *Foxd1-GFP* embryos. However, the YRF culture medium induced minimal *Foxd1*⁺ cells after 2 days of culture (Supplementary Fig. S2h). It is well known that SHH expressed in the notochord medializes the embryonic mesoderm, thereby patterning the mesoderm in a mediolateral direction^{28,29}. In contrast, BMP4 expressed in the LPM antagonizes the SHH-mediated mediolateral patterning of the mesoderm^{30,31}. In addition, *Grem1*, a BMP antagonist, was expressed in the posterior IM at E9.5, and BMP activity was lower in the *Foxd1*⁺ dorsal SPs at E11.5, as shown in Fig. 1. Indeed, addition of SHH (S) to the YRF culture media increased the sphere sizes and percentages of PDGFR α ⁺ cells in the spheres (Supplementary Fig. S2g). When

we further added the BMP receptor antagonist LDN193189 (L) to culture of the ROBO2⁺PDGFRA⁺ fraction from E9.5 *Foxd1-GFP* embryos (YRFSL condition), higher percentages of GFP⁺ cells were detected after the culture, compared with the YRF condition (Supplementary Fig. S2h). Indeed, dorsal SP genes were increased, while ventral SP genes were downregulated (Fig. 2j, Supplementary Fig. S2i). The functional competence of the cultured cells was confirmed by aggregation with E11.5 embryonic kidney-derived NPs and an isolated UB, which resulted in extensive UB branching (Fig. 2i, Supplementary Fig. S2f). Thus, the YRFSL condition is likely to represent the optimal condition for induction of dorsal SPs from the posterior IM in embryos.

Induction of dorsal SPs from mouse ESCs

Next, we examined whether mouse ESCs can be directionally differentiated into SPs. We adopted the previously reported NP induction protocol up to day 6.5¹⁴, when the cell population is likely equivalent to the posterior IM (Fig. 3a). At this stage, most of the induced cells were positive for ROBO2 (Fig. 3b). We sorted the ROBO2⁺PDGFRA⁺ fraction and cultured the cells in the SP induction condition established above. Similar to the case for cells from embryos, treatment with FGF9 (YRF) retained dorsal SP genes, while BMP4 (YRB) increased intermediate SP genes (Supplementary Fig. S3a). However, the YRF condition alone exhibited poor growth of spheres, and addition of SHH to the culture led to marked increases in viability and PDGFRA⁺ cell numbers (Supplementary Fig. S3b). We further added LDN193189 and found that expression of dorsal SP genes was increased, while expression of intermediate and ventral SP genes was decreased (Supplementary Fig. S3c). Furthermore, we compared the induced SPs (iSPs) with non-directionally induced stroma (nd-iS, Supplementary Fig. S3d): PDGFRA⁺ cells co-induced from mouse ESCs by the NP induction protocol, which employed Y27632, FGF9, and CHIR in the last step¹⁴. As expected, expression of dorsal SP genes was higher in iSPs than in nd-iS (Fig. 3c). *Aldh1a2*, which regulates UB branching, and *Fat4*, which controls NP differentiation, were expressed more abundantly in iSPs than in nd-iS (Fig. 3c). This induction protocol was also applicable to another mouse ESC line (G4-2), in which the iSPs exhibited similar gene expression features (Supplementary Fig. S3e).

scRNA-seq analyses of the iSPs showed the main *Pdgfra*⁺ iSP clusters, with a few off-target clusters that were negative for *Pdgfra* but expressed genes related to ECs, neurons, or muscles (Fig. 3d). The NP induction protocol from mouse ESCs produced ITGA8⁺PDGFRA⁻ NP clusters, PDGFRA⁺ nd-iS clusters, and a few PDGFRA⁻ (EC-like, neuron-like, or muscle-like) off-target clusters (Fig. 3d). The merged UMAP plots showed that the iSP clusters partly overlapped with the dorsal SPs in the embryonic kidney (Fig. 3d).

We next examined whether the ESC-derived iSPs satisfied the function of SPs; ability to induce

UB branching and NP differentiation. For this purpose, we isolated NPs from wild-type E11.5 embryos and UBs from E11.5 *Hoxb7-GFP* embryos, and combined them with the sorted iSPs derived from ESCs (Supplementary Fig. S3f). Reaggregation of NPs and UBs without stromal cells or with nd-iS resulted in poor branching at day 7 of culture (Fig. 3e, f). In contrast, inclusion of iSPs produced marked UB branching (Fig. 3e, f). We also confirmed that our optimized condition (YRFSL) evoked greater numbers of UB branches than the condition lacking LDN193189 (Supplementary Fig. S3g). Section *in situ* hybridization revealed expression of *Ret* in UB tips and *Wnt7b* in UB stalks of the kidney organoids containing iSPs (Fig. 3f, 2nd column), indicating that tip-stalk identity along the corticomedullary axis was established. *Six2*⁺ NPs and *Foxd1*⁺ SPs accumulated around *Ret*⁺ UB tips (Fig. 3f, 3rd column), demonstrating maintenance of NPs and SPs resulting in formation of the nephrogenic niche. Furthermore, multiple nephron cell types (glomeruli, proximal and distal renal tubules) were formed in iSP-containing organoids (Fig. 3f, 4th and 5th columns). Thus, mouse ESC-derived iSPs possess functionality to generate the complex kidney structure when combined with mouse embryo-derived UBs and NPs, by organizing UB branching and NP differentiation.

Kidney organoids completely derived from ESCs exhibit the organotypic “higher-order structure” *in vitro*

We previously reported induction protocols for NP and UB organoids^{14,17}. Our scRNA-analysis showed that the mouse ESC-derived iNP and iUB clusters overlapped with their *in vivo* counterparts (Fig. 3d). Thus, we generated the kidney structure solely derived from ESCs, by combining differentially induced NPs, UBs, and SPs (iNPs, iUBs, and iSPs). We used *Hoxb7-GFP* ESCs¹⁷ that specifically express GFP in the UB lineage to allow monitoring of UB branching in the organoids. We induced the *Hoxb7-GFP* ESCs into the three lineages using the different protocols, reaggregated the three progenitor populations, and cultured them for 7 days in the absence of growth factors (Fig. 4a). **PDGFRA⁻ off-target cells co-induced with iNPs or iSPs, as well as PDGFRA⁺ nd-iS, were likely to be eliminated by sorting based on ITGA8/PDGFR expression. The iUBs also contained stroma-like cells (Fig. 3d). However, most of these cells, which were located in the central portion of the organoids³², were likely to be surgically removed when the aggregates were prepared because we only used epithelial buds sprouting out of the UB organoids.**

Organoids **derived from the three induced progenitors** exhibited robust UB branching, with higher branching numbers than organoids generated from iNPs and iUBs alone, or iNPs, iUBs, and nd-iS (Fig. 4b, c). NPs were distributed at the periphery of the branching UBs (Fig. 4c, 2nd column, Fig. 4d), while nephrons (glomeruli, proximal and distal renal tubules) were formed inside the organoids (Fig. 4c, 3rd column, Fig. 4d). The tip-stalk identity of the UBs was

established, as shown by *Ret* and *Wnt7b* expression (Fig. 4c, 4th column), and *Six2*⁺ NPs and *Foxd1*⁺ SPs accumulated around the UB tips (Fig. 4c, 5th column). The stroma also became patterned in a corticomedullary manner, as determined by *in situ* hybridization (Fig. 4c, 6th column). A medullary stromal gene *Alx1* was clearly detected in the central region, while a cortical stromal gene *Lox* was expressed in the outer layer of the organoids. Another cortical stromal gene, *Fibin*, was expressed, albeit weakly, in the outermost layer. These expression patterns were similar to those in the embryonic kidney (Supplementary Fig. S4a) and consistent with previous scRNA-seq analyses²³. Thus, by establishing the iSP induction protocol, we successfully generated kidney organoids completely derived from ESCs with recapitulation of the “higher-order structure”: nephrons and stroma distributed along the tip-stalk orientation of branching UBs.

Multiple types of interstitial cells are differentiated in the ESC-derived kidney upon transplantation

To further promote the organoid development, we transplanted the ESC-derived kidney rudiments under the kidney capsule of immunodeficient mice. The kidney organoids with iSPs showed robust UB branching, and had grown in size (1.62 ± 0.25 mm \times 0.9 ± 0.18 mm, n=6) **at day 14 post-transplantation** (Fig. 5a). In contrast, the organoids with nd-iS showed poor UB branching (Fig. 5a) and smaller sizes (1.14 ± 0.26 mm \times 0.58 ± 0.15 mm, n=6, $p < 0.01$). In this *in vivo* setting, iUBs underwent further maturation toward collecting ducts with expression of AQP2 (principal cell marker) and CARII (intercalated cell marker) (Fig. 5a, 2nd and 3rd columns). Nephrons were aligned in a corticomedullary manner with peripherally located proximal renal tubules and the centrally located loops of Henle (Fig. 5a, fourth column). A medullary stromal gene *Alx1* was expressed in central regions of the organoids, and the innermost region expressed *Wnt4* in a partly overlapping manner with *Alx1* (Fig. 5b, 1st and 2nd columns), which was similar to the situation *in vivo* (Supplementary Fig. S4a). These observations indicate formation of the deeper medullary stroma, and are consistent with a previous report on *Wnt4* expression³³. In contrast, organoids with nd-iS exhibited poorly patterned nephrons, UBs/collecting ducts, and stroma (Fig. 5a, b).

We examined the development of glomeruli in more detail, because SPs *in vivo* are known to differentiate into specialized stromal cells: mesangial cells, which form inside the glomeruli, and renin cells, which surround the afferent arterioles coming into the glomeruli^{7,34}. As we previously reported³⁵, glomeruli in the transplants were well vascularized with PECAM1⁺ ECs and the slit diaphragm-related protein NEPHRIN was distributed in a linear manner on the basal side (facing the ECs) of the podocytes (Fig. 5b, 3rd and 4th columns). Inside these glomeruli, we detected cells positive for HOPX, a representative mesangial cell marker. The percentages of HOPX⁺ glomeruli in organoids with iSPs were significantly higher than those in organoids with nd-iS (Fig. 5c). Angiotensin receptor type 1a (*Agtr1a*), another mesangial cell marker, was detected

inside the glomeruli (Fig. 5d, 1st column), and *Ren1* (mouse *renin* gene) was detected at sites close to the glomeruli. In addition, *Ehd3*⁺ glomerular capillaries, as well as *Gja5*⁺ periglomerular arterioles and medullary arteries, were formed (Supplementary Fig. S4b), suggesting the establishment of functional vascularization in the organoids.

We also detected ureter-like structures, consisting of epithelia expressing UPK1B (urothelial marker) surrounded by stroma expressing MYH11 (ureteric stromal marker) (Fig. 5d, 2nd column). When harvested at a later timepoint (day 20 post-transplantation), marked dilatation of the ureter was observed (Supplementary Fig. S4c), suggesting possible urine accumulation in the ureter. Although we aimed to induce dorsal SPs, the expression levels of some intermediate and ventral SP genes remained higher than those *in vivo* (Fig. 3c, Supplementary Fig. 3e). This may have enabled iSPs to differentiate into the ureteric stroma at the UB stalk, thereby inducing the iUB epithelia to differentiate into UPK1A⁺ urothelial cells. Thus, the ESC-derived iSPs differentiated into cortical and medullary stroma, intraglomerular mesangial cells, extraglomerular renin cells, and even smooth muscle cells around the ureter.

Nephron, UB, and stromal lineages in the ESC-derived transplanted kidney express perinatal renal genes

We performed scRNA-seq analysis of the transplanted ESC-derived kidney (day 14 post-transplantation) and compared the data with those for embryonic kidneys during late gestation described in our previous report³⁶. The analysis demonstrated the presence of multiple types of nephrons, UBs, and stroma in organoids generated using iSPs (Fig. 6a). The organoids using nd-iS also had complex clusters judging from the poor histological structures shown in Fig. 5, but the individual clusters were not as discrete as those in the iSP-organoids. In both types of organoids, we confirmed the ESC origin of all three renal lineages based on the sex difference between the donor ESCs (male) and the host animals (female). Female-specific *Xist* was undetectable in nephrons, UBs/collecting ducts, and stromal cells, but was detected in ECs, lymphocytes, and leukocytes (Fig. 6b). This is consistent with our previous finding that the host-derived vasculature integrates into organoids upon transplantation³⁵. Apparent off-target clusters, such as neurons or muscles, were not detected (Fig. 6a), presumably because we purified the progenitors before aggregation.

The parenchymal clusters in iSP-derived organoids overlapped with the clusters in the E15.5 and neonatal (P0) kidney *in vivo* (Fig. 6a). The nephron clusters for glomerular podocytes, proximal and renal tubules, loops of Henle, and distal tubules, as well as the UB/collecting duct clusters of principal cells and intercalated cells, were clearly detected (Fig. 6a, Supplementary Fig. S5a). Maturation markers detectable at P0 but not E15.5³⁶, such as those for podocytes (*Col4a3*, *Htra1*), proximal tubules (*Slc5a2*, *Fbp1*), principal cells (*Aqp2*), and intercalated cells

(*Atp6vg3*), were also expressed in the iSP-derived organoids (Supplementary Fig. S5a). In contrast, NPs (*Six2*⁺) and UB tips (*Ret*⁺) were rarely detected (Fig. 6a, arrowheads, Supplementary Fig. S5a), indicating depletion of the nephrogenic niche. This niche depletion was not specific to organoids, because it was also observed upon transplantation of embryonic kidneys³⁶, although the precise mechanism remains unclear. Glomerular ECs (*Ehd3*⁺) and periglomerular arterioles (*Gja5*⁺) were also detected, but expression of some representative genes was lower than that in glomerular ECs *in vivo* (Supplementary Fig. S4d). Because the ECs were not derived from mouse ESCs, these data imply the flexibility, as well as the limitation, of the adult ECs in the host animals.

Stromal cell clusters from the scRNA-seq analysis were extracted for further detailed analysis (Fig. 6c–d). The iSP-derived organoids showed mostly overlapping clusters with those *in vivo* (P0), including the cortical and medullary stroma, mesangial cells, renin cells, and smooth muscle cells around the ureter (Fig. 6c, Supplementary Fig. S5b). In contrast, in the organoids derived from nd-iS, only a small fraction of the cells contributed to the authentic cell types, and 43.9% of the cells (1152 cells in Clusters 0 and 9 out of 2626 cells in total) were positioned distantly from those *in vivo* (Fig. 6c). We also detected an ectopic cluster in the iSP-derived organoids (Fig. 6c), but with a smaller percentage (2.8%; 81 cells in Cluster 35 out of 2879 cells in total). An unbiased hierarchy clustering analysis also showed that the iSP-derived interstitial cells were similar to those *in vivo* than the nd-iS-derived interstitial cells (Fig. 6d, Supplementary Data2). We selected representative markers for each stromal subpopulation *in vivo* (P0), and examined their expression levels in the organoids (Supplementary Data1). UMAP and dot plot analysis showed that many stromal marker genes in the organoids were expressed at comparable levels to P0 (Fig. 6e, Supplementary Fig. S5b). Expression of SP-related genes was decreased, consistent with the nephrogenic niche depletion described above. Many stromal markers appeared to be expressed in the nd-iS-derived organoids but, as described above, only a small fraction of the cells contributed to the authentic cell types, and thus a small number of cells managed to differentiate despite the poor three-dimensional structure. The organoid-specific clusters expressed some ectopic genes, including *Pgr* and *Ngfr* (Fig. 6e, Supplementary Fig. S5b, and Supplementary Data1). Taken together, ESC-derived kidney organoids generated by combining iNPs, iUBs, and iSPs, were organized with multiple types of stroma and parenchyma (nephrons and UBs/collecting ducts), and expressed many genes detected in neonatal kidneys.

Discussion

By elucidating the molecular features of the *in vivo* renal stromal lineages at single-cell resolution, we established *in vitro* protocols to induce dorsal SPs from mouse ESCs. When ESC-derived SPs, NPs, and UBs were assembled, the organotypic kidney structure that consists of branching UBs with multiple nephrons was generated. Furthermore, the iSPs could differentiate into multiple

types of stroma: cortical and medullary stromal cells, as well as mesangial cells and renin cells.

Our data revealed the importance of mediolateral and dorsoventral patterning in kidney specification. Our SP induction protocol involves SHH and a BMP inhibitor. Although the role of SHH in early embryogenesis was reported²⁸⁻³⁰, the effect of SHH on the kidney lineage precursors, namely the IM, has remained unclear. SHH expressed in the midline medializes the mesoderm, while BMP in the LPM lateralizes the mesoderm. Because the IM is formed between the LPM and the PM (close to the midline), the balance between SHH and BMP is likely to be critical. Indeed, the IM expressed BMP antagonist *Grem1*, and manipulation of SHH and BMP signals enhanced SP induction *in vitro*. Our data further suggest that dorsoventral patterning is important for specifying the three types of SPs: FGF induced dorsal SPs, while BMP enhanced ventral gene expression. Indeed, these patterning concepts enabled generation of dorsal SPs that exhibited similar gene expression profiles to their *in vivo* counterparts, as well as functionality, in sharp contrast to the non-parenchymal cells (nd-iS) in the conventional NP organoids.

When dorsal iSPs were combined with iNPs and iUBs, kidney tissue with the organotypic structure was obtained purely from mouse ESCs. This emphasizes the importance of stroma for generation of the “higher-order structure” of the organ. As shown in our previous report¹⁷, assembly of differentially induced progenitors is an effective approach to generate the organotypic structure, although a single induction protocol is frequently adopted in the organoid research field. Several groups recently reaggregated differentially induced human nephron organoids (theoretically containing NPs and nd-iS) and UB organoids and showed some mutual interactions^{3,19,20}. However, properly branched UBs were not formed. Our mouse kidney organoids contained branching UBs in the center and multiple types of nephrons at the periphery, which was only achievable in the presence of authentic stroma. Although the precise interaction mechanisms between the stroma and parenchyma remain to be elucidated, previous loss-of-function studies *in vivo* have revealed some of the responsible genes/signals from stroma to parenchyma, including *Gdnf* and *Aldh1a2* for UBs, and *Fat4* for NPs^{5,11,12}. Indeed, iSPs expressed many of these genes at high levels. Our building-up strategy using organoids will contribute to identification of unknown molecules involved in the stroma-parenchymal interactions. **ECs are not primarily required for the initial formation of the higher-order structure, because ECs were excluded from the PDGFRA⁺ SP fraction before aggregation¹⁷. However, upon transplantation, ECs are likely to contribute to the maturation of stromal and parenchymal cells in glomeruli, and possibly to the corticomedullary patterning of the renal stroma in concert with UBs.**

The present data demonstrate that iSPs can differentiate into multiple types of the stroma. The corticomedullary axis was formed in our organoids, and at least four layers of the stroma (*Fibin*⁺ outermost cortical stroma, *Lox*⁺ cortical stroma, *Alx1*⁺/*Wnt4*⁻ medullary stroma, and *Alx1*⁺/*Wnt4*⁺ deeper medullary stroma) were generated. It was reported that *Wnt7b* in the UB stalks emits its

signal to the surrounding medullary stroma and is required for renal medulla formation and nephron elongation in the medulla³⁷. Our data suggest that such interactions between stroma and parenchyma occurred in the organoids. Formation of mesangial cells and renin cells was only observed upon transplantation, suggesting that the *in vitro* culture condition was not optimal for stromal differentiation. After transplantation, the glomeruli became vascularized and specialized stromal cells were formed, probably through interactions between SPs, ECs, and glomerular podocytes. Renin is a secreted hormone produced by the juxtaglomerular apparatus that cleaves angiotensinogen in the plasma to generate angiotensin I³⁸. Subsequent processing generates angiotensin II, which binds to its receptors, including AGTR1A expressed in mesangial cells. Mesangial cells contract in response to angiotensin II, thereby regulating glomerular filtration through glomerular capillaries³⁹. Thus, the generation of renin cells and mesangial cells is a significant step toward functionality of the kidney. Recently, cAMP stimulation of conventional kidney organoids was shown to induce renin production⁴⁰. Combination of our protocol with cAMP stimulation may further increase the induction efficiency for renin cells.

There remain several challenges toward functional kidney generation. Mouse kidney organoids lack an elongated ureter and remain unconnected to the host urinary outlet; thus, a complete urinary flow route is not established. Because we now know that addition of BMP4 favors intermediate and ventral SP formation, it will become achievable to generate more elongated ureters with smooth muscle layers, which are descendants of *Tbx18*⁺ intermediate SPs. It is also important to examine whether our SP induction protocol can be applicable to human iPSCs. **The similarity in gene expression between mouse and human SPs remains unclear. For example, *FOXD1* was reported to be expressed in SPs and NPs at almost equal levels in humans⁴¹. Accumulating scRNA-seq data on human embryonic kidneys^{41,42} will be valuable for identification of SP markers in humans, and for precise assessment of the induced SPs and resultant stroma. For the higher-order kidney structure in humans, the quality of the induced NPs and UBs also needs to be fine-tuned based on information obtained *in vivo*.**

Taken together, we have revealed the developmental program of renal SPs and successfully induced these cells *in vitro*. Inclusion of iPSCs enabled the generation of organoids with the organotypic structure of the kidney equipped with multiple types of stromal cells. Our study provides a proof-of-concept for the generation of organoids with the “higher-order structure” derived purely from PSCs. It also serves as a useful platform to study the heterogeneous stromal cells in organs.

Methods

Mice

Osr1-GFP mice and *Tbx18-MerCreMer* mice were generated as described previously^{14,43}. *Hoxb7-*

GFP mice⁴⁴, *Foxd1-GFP* mice⁷, *Foxd1-GFP*CreER^{T2} mice⁷, *Isl1-MerCreMer* mice⁴⁵, and *ROSA26-CAG-tdTomato* mice⁴⁶ were purchased from Jackson Laboratory. *Foxd1-GFP*CreER^{T2}, *Isl1-MerCreMer*, and *Tbx18-MerCreMer* mice were maintained on a C57BL/6 background, while the other strains were maintained on a mixed genetic background. For transplantation experiments, 8–10-week-old immunodeficient male mice (NOD/ShiJic-scidJcl) were purchased from Charles River Laboratory Japan Inc. The mice were housed in a specific pathogen-free animal facility in plastic cages on a 12-h/12-h light/dark cycle, and fed an irradiated CE-2 diet (CLEA Japan Inc.). All animal experiments were performed in accordance with institutional ethical guidelines and approved by the licensing committee of Kumamoto University (approval number: A2019–113).

SP induction from mouse ESCs

The *Hoxb7-GFP* mouse ESC line (clone #B6-5; male) was described previously¹⁷ and maintained on mitotically inactivated murine embryonic fibroblasts. The G4-2 mouse ESC line (male) with ubiquitous expression of GFP⁴⁷ was kindly provided by Dr. Hitoshi Niwa (Kumamoto University) and cultured on gelatin-coated plates. Dissociated mouse ESCs were aggregated and induced using the previously described NP induction protocol up to day 6.5^{14,17}. The spheres were dissociated with Accutase at 37°C for 8 min and the ROBO2⁺PDGFRA⁺ fraction was sorted. The sorted cells were resuspended in differentiation medium (20,000 cells/ 200 μ l) in the presence of Y-28632 (10 μ M), RA (0.1 μ M), FGF9 (10 ng/ml), SHH-N (1 μ g/ml), and LDN193189 (25 nM), seeded into 96-well low-cell-binding U-bottom plates (Thermo), and cultured until day 9.5. The composition of the differentiation medium was 75% IMDM (Gibco) and 25% Ham's F12 medium (Gibco) supplemented with 0.5 \times B27 without RA (Gibco), 0.5 \times N2 (Gibco), 0.05% BSA (Sigma), 0.5 mM ascorbic acid (Sigma), 1 \times GlutaMAX (Gibco), 0.5 \times penicillin/streptomycin (Gibco), and 4.5 \times 10⁻⁴ M 1-thioglycerol (Sigma).

Kidney reconstitution by combining three types of progenitors

NPs, UBs, and SPs were isolated from E11.5 mouse kidneys or induced from mouse ESCs. NPs and SPs were sorted based on expression of ITGA8 and PDGFRA, and UBs were isolated as intact buds as described previously¹⁷. To create a kidney organoid with the “higher-order structure”, 50,000 SPs and 25,000 NPs were mixed and seeded into 96-well low-cell-binding U-bottom plates (Thermo #174925), followed by centrifugation (210 \times g, 4 min). Single UBs isolated manually using tungsten needles were placed onto the deposited sheet-like cells. After culture at 37°C overnight, the aggregated spheroids were transferred to Transwell inserts (Costar) containing 50 μ l of 50% Matrigel (growth factor reduced; Corning) in DMEM/F12 with 10% FBS, and cultured for 7 days for *in vitro* analyses. The organoids were transplanted at day 3.

Data Availability

All data supporting the conclusions are present in the paper and the supplementary materials. A reporting summary for this article is available as a Supplementary Information file. The scRNA-seq data have been deposited in the National Center for Biotechnology Information Gene Expression Omnibus (GSE178263). The scRNA-seq data of E15.5 and P0 embryonic kidneys were described previously (GSE149134)³⁶. Source data are provided with this paper.

References

1. Nishinakamura, R. Human kidney organoids: progress and remaining challenges. *Nat. Rev. Nephrol.* **15**, 613–624 (2019).
2. Combes, A. N. *et al.* Single cell analysis of the developing mouse kidney provides deeper insight into marker gene expression and ligand-receptor crosstalk. *Development* **146**, dev178673 (2019).
3. England, A. R. *et al.* Identification and characterization of cellular heterogeneity within the developing renal interstitium. *Development* **147**, dev190108 (2020).
4. Han, L. *et al.* Single cell transcriptomics identifies a signaling network coordinating endoderm and mesoderm diversification during foregut organogenesis. *Nat Commun* **11**, 4158 (2020).
5. Costantini, F. & Kopan, R. Patterning a complex organ: branching morphogenesis and nephron segmentation in kidney development. *Dev Cell* **18**, 698–712 (2010).
6. Kobayashi, A. *et al.* *Six2* defines and regulates a multipotent self-renewing nephron progenitor population throughout mammalian kidney development. *Cell Stem Cell* **3**, 169–181 (2008).
7. Kobayashi, A. *et al.* Identification of a multipotent self-renewing stromal progenitor population during mammalian kidney organogenesis. *Stem Cell Reports* **3**, 650–662 (2014).
8. Buechler, M. B. *et al.* Cross-tissue organization of the fibroblast lineage. *Nature* **593**, 575–579 (2021).
9. Carroll, T. J., Park, J.-S., Hayashi, S., Majumdar, A. & McMahon, A. P. Wnt9b plays a central role in the regulation of mesenchymal to epithelial transitions underlying organogenesis of the mammalian urogenital system. *Dev Cell* **9**, 283–292 (2005).
10. Barak, H. *et al.* FGF9 and FGF20 Maintain the Stemness of Nephron Progenitors in Mice and Man. *Dev. Cell* **22**, 1191–1207 (2012).
11. Rosselot, C. *et al.* Non-cell-autonomous retinoid signaling is crucial for renal development. *Development* **137**, 283–292 (2010).
12. Das, A. *et al.* Stromal-epithelial crosstalk regulates kidney progenitor cell differentiation. *Nat Cell Biol* **15**, 1035–1044 (2013).
13. Sequeira-Lopez, M. L. S. *et al.* The earliest metanephric arteriolar progenitors and their role in

- kidney vascular development. *Am. J. Physiol. Integr. Comp. Physiol.* **308**, R138–R149 (2015).
14. Taguchi, A. *et al.* Redefining the in vivo origin of metanephric nephron progenitors enables generation of complex kidney structures from pluripotent stem cells. *Cell Stem Cell* **14**, 53–67 (2014).
 15. Morizane, R. *et al.* Nephron organoids derived from human pluripotent stem cells model kidney development and injury. *Nat. Biotechnol.* **33**, 1193–1200 (2015).
 16. Takasato, M. *et al.* Kidney organoids from human iPS cells contain multiple lineages and model human nephrogenesis. *Nature* **526**, 564–568 (2015).
 17. Taguchi, A. & Nishinakamura, R. Higher-order kidney organogenesis from pluripotent stem cells. *Cell Stem Cell* **21**, 730–746 (2017).
 18. Fetting, J. L. *et al.* FOXD1 promotes nephron progenitor differentiation by repressing decorin in the embryonic kidney. *Development* **141**, 17–27 (2014).
 19. Tsujimoto, H. *et al.* A Modular Differentiation System Maps Multiple Human Kidney Lineages from Pluripotent Stem Cells. *Cell Rep.* **31**, 107476 (2020).
 20. Uchimura, K., Wu, H., Yoshimura, Y. & Humphreys, B. D. Human pluripotent stem cell-derived kidney organoids with improved collecting duct maturation and injury modeling. *Cell Rep.* **33**, 108514 (2020).
 21. Howden, S. E. *et al.* Plasticity of distal nephron epithelia from human kidney organoids enables the induction of ureteric tip and stalk. *Cell Stem Cell* **28**, 671–684.e6 (2021).
 22. Mugford, J. W., Sipilä, P., McMahon, J. a & McMahon, A. P. Osr1 expression demarcates a multi-potent population of intermediate mesoderm that undergoes progressive restriction to an Osr1-dependent nephron progenitor compartment within the mammalian kidney. *Dev Biol* **324**, 88–98 (2008).
 23. Bohnenpoll, T. *et al.* Tbx18 expression demarcates multipotent precursor populations in the developing urogenital system but is exclusively required within the ureteric mesenchymal lineage to suppress a renal stromal fate. *Dev. Biol.* **380**, 25–36 (2013).
 24. Yang, L. *et al.* Isl1Cre reveals a common Bmp pathway in heart and limb development. *Development* **133**, 1575–85 (2006).
 25. Miyazaki, Y., Oshima, K., Fogo, A., Hogan, B. L. & Ichikawa, I. Bone morphogenetic protein 4 regulates the budding site and elongation of the mouse ureter. *J. Clin. Invest.* **105**, 863–873 (2000).
 26. Pearce, J. J. H., Penny, G. & Rossant, J. A mouse cerberus/Dan-related gene family. *Dev Biol* **209**, 98–110 (1999).
 27. Michos, O. *et al.* Gremlin-mediated BMP antagonism induces the epithelial-mesenchymal feedback signaling controlling metanephric kidney and limb organogenesis. *Development* **131**, 3401–3410 (2004).

28. Johnson, R. L., Laufer, E., Riddle, R. D. & Tabin, C. Ectopic expression of Sonic hedgehog alters dorsal-ventral patterning of somites. *Cell* **79**, 1165–1173 (1994).
29. Fan, C. M. & Tessier-Lavigne, M. Patterning of mammalian somites by surface ectoderm and notochord: Evidence for sclerotome induction by a hedgehog homolog. *Cell* **79**, 1175–1186 (1994).
30. Pourquié, O. *et al.* Lateral and axial signals involved in avian somite patterning: A role for BMP4. *Cell* **84**, 461–471 (1996).
31. Tonegawa, A., Funayama, N., Ueno, N. & Takahashi, Y. Mesodermal subdivision along the mediolateral axis in chicken controlled by different concentrations of BMP-4. *Development* **124**, 1975–1984 (1997).
32. Kuraoka, S. *et al.* PKD1-dependent renal cystogenesis in human induced pluripotent stem cell-derived ureteric bud/collecting duct organoids. *J Am Soc Nephrol* **31**, 2355–2371 (2020).
33. Itäranta, P. *et al.* Wnt-4 signaling is involved in the control of smooth muscle cell fate via Bmp-4 in the medullary stroma of the developing kidney. *Dev. Biol.* **293**, 473–483 (2006).
34. Humphreys, B. D. *et al.* Fate tracing reveals the pericyte and not epithelial origin of myofibroblasts in kidney fibrosis. *Am J Pathol* **176**, 85–97 (2010).
35. Sharmin, S. *et al.* Human induced pluripotent stem cell-derived podocytes mature into vascularized glomeruli upon experimental transplantation. *J. Am. Soc. Nephrol.* **27**, 1778–1791 (2016).
36. Naganuma, H. *et al.* Molecular detection of maturation stages in the developing kidney. *Dev Biol* **470**, 62–73 (2021).
37. Yu, J. *et al.* A Wnt7b-dependent pathway regulates the orientation of epithelial cell division and establishes the cortico-medullary axis of the mammalian kidney. *Development* **136**, 161–171 (2009).
38. Brunskill, E. W. *et al.* Genes that confer the identity of the renin cell. *J Am Soc Nephrol* **22**, 2213–25 (2011).
39. Stockand, J. D. & Sansom, S. C. Glomerular mesangial cells: Electrophysiology and regulation of contraction. *Physiol. Rev.* **78**, 723–744 (1998).
40. Shankar, A. S. *et al.* Human kidney organoids produce functional renin. *Kidney Int* **99**, 134–147 (2021).
41. Lindström, N. O. *et al.* Conserved and divergent features of mesenchymal progenitor cell types within the cortical nephrogenic niche of the human and mouse kidney. *J Am Soc Nephrol* **29**, 806–824 (2018).
42. Wang, P. *et al.* Dissecting the global dynamic molecular profiles of human fetal kidney development by single-cell RNA sequencing. *Cell Rep.* **24**, 3554–3567.e3 (2018).
43. Grisanti, L. *et al.* Tbx18 targets dermal condensates for labeling, isolation, and gene ablation

- during embryonic hair follicle formation. *J. Invest. Dermatol.* **133**, 344–353 (2013).
44. Yu, J., Carroll, T. J. & McMahon, A. P. Sonic hedgehog regulates proliferation and differentiation of mesenchymal cells in the mouse metanephric kidney. *Development* **129**, 5301–5312 (2002).
 45. Laugwitz, K.-L. *et al.* Postnatal *Isl1*⁺ cardioblasts enter fully differentiated cardiomyocyte lineages. *Nature* **433**, 647–53 (2005).
 46. Madisen, L. *et al.* A robust and high-throughput Cre reporting and characterization system for the whole mouse brain. *Nat Neurosci* **13**, 133–140 (2010).
 47. Niwa, H., Miyazaki, J. I. & Smith, A. G. Quantitative expression of Oct-3/4 defines differentiation, dedifferentiation or self-renewal of ES cells. *Nat. Genet.* **24**, 372–376 (2000).

Acknowledgments

We thank S. Fujimura, S. Usuki, M. Yamane, T. Ichikawa, K. Yasunaga, H. Naganuma, and S. Kuraoka for experimental assistance. We also thank Alison Sherwin, PhD, from Edanz Group (<https://jp.edanz.com/ac>) for editing a draft of this manuscript. This work was supported in part by a KAKENHI Grant (JP21H05050) from the Japan Society for the Promotion of Science and the Research Center Network for Realization of Regenerative Medicine (21bm0804013h0005) from the Japan Agency for Medical Research and Development

Author contributions

Conceptualization, R.N. and A.T.; Methodology, S.T., E.T., and A.T.; Investigation, S.T., E.T., K.M., T.O., D.I. and A.K.; Resources, R.N., and CL.C.; Writing – Original Draft, S.T., E.T., and R.N.; Writing – Review & Editing, A.K., A.T., and R.N.; Funding Acquisition, R.N.; Project Administration and Supervision, R.N.

Competing interests

The authors declare no competing interests.

Figure legends

Figure 1. RA, FGF, and BMP signaling regulates dorsoventral patterning of renal SPs

(a) Three domains of SPs, shown by immunostaining of the kidney at E11.5. Left panel: FOXD1⁺ dorsal SPs (green); middle panel: TBX18⁺ intermediate SPs (green); right panel: ISL1⁺ ventral SPs (magenta). *SIX2* and *KRT8* are also stained to mark NPs and UBs, respectively. Scale bars: 15 μ m.

(b–e) scRNA-seq analysis of the stromal cells in the mouse E11.5 kidney. (b) UMAP plots

showing **five clusters, with two representing *Top2a*⁺ proliferating cells**. (c, d) Representative genes in dorsal, intermediate, and ventral SPs, shown as UMAP plots (c) and dot plots (d). (e) UMAP plots showing signal-related genes.

(f) Schematic diagram of isolation of *Osr1-GFP*⁺ cells from E9.5 embryos, followed by a 2-day culture.

(g–i) Expression of stromal domain-related genes after the culture, analyzed by qRT-PCR. (g) RA induces dorsal SP-related genes (n=5). (h, i) RA and FGF9 induce dorsal SP-related genes, while RA and BMP4 induce intermediate and ventral SP-related genes (n=3). Relative mRNA expression levels normalized to β -actin gene expression are shown as mean \pm SEM. Student's *t*-test was performed in (g) and Dunnett's multiple comparison test was performed in (h, i). **The source data are provided as a Source Data file.** F+: Foxd1-GFP⁺PDGFRA⁺ dorsal SPs harvested at E11.5 (non-cultured; presented as a reference); Y: Y27632 (10 nM); R: RA (0.1 μ M); F1: FGF9 (1 ng/ml); F10: FGF9 (10 ng/ml); F30: FGF9 (30 ng/ml); B0.1: BMP4 (0.1 ng/ml); B1: BMP4 (1 ng/ml); B10: BMP4 (10 ng/ml).

Figure 2. ROBO2⁺/PDGFRA⁺ IM is induced toward dorsal SPs *in vitro*

(a, b) scRNA-seq analysis of the posterior part (caudal from the 26th somite) of the E9.5 mouse embryo. (a) UMAP plots showing multiple clusters including the IM. (b) Representative genes for the IM. Note the overlap of *Osr1*, *Robo2*, and *Grem1*. Black arrowheads: IM; white arrowheads: LPM; arrows: neural tube (NT). NC: neural crest; HL: hindlimb bud; WD: Wolffian duct.

(c) *In situ* hybridization of *Osr1*, *Robo2*, and *Grem1* in the posterior IM at E9.5. The dorsal side faces upward, while the ventral side faces downward. Left panels: *Osr1* and *Robo2* overlap in the IM (arrowhead) and LPM (arrow). Right panels: *Grem1* is expressed in the IM (arrowhead). Scale bars: 50 μ m.

(d) Schematic diagram for isolation of ROBO2⁺PDGFRA⁺ cells from wild-type or *Osr1-GFP* embryos at E9.5, followed by a 2-day culture under the YRF condition (Y: Y27632 [10 nM]; R: RA [0.1 μ M]; F: FGF9 [10 ng/ml]).

(e) Flow cytometric analysis before the culture (day 0). Most of the ROBO2⁺PDGFRA⁺ cells (red) are *Osr1-GFP*^{high}, while ROBO2⁻PDGFRA⁺ cells (green) and PDGFRA⁻ cells (blue and magenta) exhibit varying degrees of *Osr1-GFP* expression.

(f) Expression of IM-related genes in the indicated cell fractions before the culture. The genes are enriched in ROBO2⁺PDGFRA⁺ cells. G+: *Osr1-GFP*⁺ cells; G-: *Osr1-GFP*⁻ cells; R-P+: ROBO2⁻PDGFRA⁺ cells; R+P+: ROBO2⁺ PDGFRA⁺ cells. Data are shown as mean \pm SEM (n=3). Student's *t*-test was performed.

(g) Spheres after the culture. Upper panels: Photos of spheres from the indicated fractions. Scale

bars: 100 μm . Right graph: Diameters of the spheres. Data are shown as mean \pm SEM (n=17, 36, and 31, respectively). The Tukey–Kramer test was performed. Lower panels: flowcytometric analysis of the induced spheres.

(h) Expression of stromal domain-related genes after the culture. F+: Foxd1-GFP⁺PDGFRA⁺ dorsal SPs harvested at E11.5 (non-cultured; presented as a reference). G+: *Osr1-GFP*⁺ cells; R–P+: ROBO2[–] PDGFRA⁺ cells; R+P+: ROBO2⁺PDGFRA⁺ cells.

Data are shown as mean \pm SEM (n=3). Student’s *t*-test was performed.

(i) Aggregation assay for UB branching. The ROBO2⁺ PDGFRA⁺ (R+) fractions were cultured for 2 days in the indicated conditions, combined with E11.5 embryo-derived NPs and UBs (tdTomato⁺), and cultured for a further 7 days. Scale bars: 200 μm . PDGFRA⁺: freshly isolated stromal cells from E11.5 embryos used as a reference for re-aggregation.

(j) Expression of genes related to dorsoventral patterning in GFP⁺ cells cultured from the ROBO2⁺/PDGFRA⁺ fraction of E9.5 Foxd1-GFP embryos. Data are shown as mean \pm SEM (n=3). Student’s *t*-test was performed.

(f–h, j) The source data are provided as a Source Data file.

Figure 3. Mouse ESC-derived SPs support the generation of the complex kidney structure

(a) SP induction protocol from mouse ESCs. A: activin (10 ng/ml); B: BMP4 (3 ng/ml); C10: CHIR (10 μM); C3: CHIR (3 μM); R: RA (0.1 μM); F: FGF9 (10 ng/ml).

(b) Flow cytometric analysis of the induced cells at day 6.5 and 9.5.

(c) Comparison of expression levels of stromal domain-related genes between nd-iS and iSPs. F+: Foxd1-GFP⁺PDGFRA⁺ dorsal SPs harvested at E11.5 (non-cultured; presented as a reference). Data are shown as mean \pm SEM (n=3). Student’s *t*-test was performed. The source data are provided as a Source Data file.

(d) scRNA-seq analysis of mouse ESC-derived progenitors (iSPs, iNPs, iUBs) and E13.5 embryonic kidney. UMAP plots are shown. Arrows: dorsal SPs of the embryonic kidney. Arrowhead: nd-iS co-induced with iNP. EC: endothelial cells; Leu: leucocytes; iUB-S: stroma co-induced with iUB; *: dying cells.

(e) UB branch numbers in the aggregates. Data are shown as mean \pm SEM (n=7, 7, and 8, respectively). The Tukey-Kramer test was performed. The source data are provided as a Source Data file.

(f) Aggregates formed without stromal cells (–), with nd-iS, or with iSPs. NPs and UBs are isolated from the E11.5 embryonic kidneys. 1st column: GFP images of UBs; 2nd column: *in situ* hybridization of *Wnt7* (UB stalks) and *Ret* (UB tips); 3rd column: *in situ* hybridization of *Six2* (NPs), *Foxd1* (SPs), and *Ret* (UB tips); 4th column: immunostaining of KRT8 (UBs), CDH1 (UBs,

distal tubules), and LTL (proximal tubules); 5th column: immunostaining of KRT8 (UBs) and NPHS1 (glomerular podocytes). Scale bars: 100 μ m.

Figure 4. Kidney organoids solely derived from ESCs exhibit the organotypic “higher-order structure” *in vitro*

(a) Schematic diagram of kidney organoid formation. iSPs, iNPs, and iUBs derived from *Hoxb7-GFP* mouse ESCs are combined, and cultured at the air/liquid interface for 7 days.

(b) UB branch numbers in kidney organoids. Data are shown as mean \pm SEM (n=16, 15, and 18, respectively). The Tukey–Kramer test was performed. **The source data are provided as a Source Data file.**

(c) Mouse ESC-derived organoids formed without stromal cells (–), with nd-iS, or with iSPs. 1st column: GFP images of UBs; 2nd column: whole-mount immunostaining of SIX2 (NPs) and KRT8 (UBs); 3rd column: whole-mount immunostaining of KRT8 (UBs), NPHS1 (glomerular podocytes), LTL (proximal tubules), and CDH1 (UBs, distal tubules); 4th column: *in situ* hybridization of *Wnt7* (UB stalks) and *Ret* (UB tips); 5th column: *in situ* hybridization of *Six2* (NPs), *Foxd1* (SPs), and *Ret* (UB tips); 6th column: *in situ* hybridization of *Alx1* (medullary stroma), *Lox* (cortical stroma), and *Fibin* (outer layer of cortical stroma). Scale bars: 100 μ m.

(d) Digitized images of the whole-mount staining data in the 2nd and 3rd columns of (C). Scale bars: 100 μ m.

Figure 5. Multiple types of interstitial cells are differentiated in the ESC-derived kidney upon transplantation

(a) Differentiation of UBs and nephrons in mouse ESC-derived transplanted organoids (generated using iSPs or nd-iS). 1st column: GFP images of UBs; 2nd column: *in situ* hybridization of *Wnt7b* (UB stalks) and *Aqp2* (principal cells); 3rd column: immunostaining of CAR2 (intercalated cells), AQP2 (principal cells), and KRT8 (UBs); 4th column: immunostaining of KRT8 (UBs), SLC12A1 (loops of Henle), and LTL (proximal tubules). Scale bars: columns 1–3, 50 μ m; column 4, 100 μ m.

(b) Differentiation of stromal cells. 1st column: *in situ* hybridization of *Alx1* (medullary stroma) and *Wnt4* (innermost medullary stroma); 2nd column: magnified images of the first columns; 3rd column: immunostaining of HOPX (mesangial cells), PECAM1 (ECs), and NPHS1 (glomerular podocytes); 4th column: magnified images of the 3rd columns. Scale bars: 100 μ m.

(c) Percentages of glomeruli equipped with HOPX⁺ mesangial cells. Data are shown as mean \pm SEM (n=6 per group). The non-parametric Mann–Whitney U test was performed. **The source data are provided as a Source Data file.**

(d) Differentiation of mesangial cells and ureteric stroma. 1st column: *in situ* hybridization of

Ren1 (renin cells), *Agtr1* (mesangial cells), and *Nphs1* (podocytes); 2nd column: staining of MYH11 (ureteric stroma), UPK1B (ureter epithelium), and KRT8 (UBs). Scale bars: 50 μ m.

All of the data were obtained at day 14 post-transplantation, except for the GFP images of UBs obtained at day 10 post-transplantation.

Figure 6. Nephron, UB, and stromal lineages in the ESC-derived kidney express perinatal renal genes

(a) UMAP plots of embryonic kidneys (E15.5 and P0) and mouse ESC-derived transplanted organoids (generated using iSPs or nd-iS). The lack of NPs and UB tips in the transplanted organoids is indicated by red and black arrows, respectively. Pod: podocytes; PEC: parietal epithelial cells; PT: proximal tubule; LoH: loop of Henle; DT: distal tubule; PC: principal cells; IC: intercalated cells; UE: uroepithelium; Ren: renin cells; Mes: mesangial cells; CS: cortical stroma; oMS: outer medullary stroma; iMS: inner medullary stroma; US: ureteric stroma; Leu: leukocytes; Lym: lymphocytes; *: organoid-specific clusters; #: proliferating cells.

(b) UMAP plots of *Xist*. *Xist* is absent in nephrons, UBs, and stroma derived from male mouse ESCs, but detected in the female host-derived lymphocytes (Lym), leukocytes (Leu), and ECs.

(c) UMAP plots of extracted stromal cells in the P0 kidney and organoids. Mes: mesangial cells; Ren: renin cells; CS: cortical stroma; oMS: outer medullary stroma; iMS: inner medullary stroma; US: ureteric stroma; SCS: subcapsular stroma; *: organoid-specific clusters; #: proliferating cells.

(d) Unbiased hierarchical clustering analysis of the induced stroma and embryonic stroma.

(e) UMAP plots of representative genes in the stromal cells of the embryonic kidneys (E15.5 and P0), iSP-derived organoids, and nd-iS-derived organoids. Ren: renin cells; Mes: mesangial cells; CS: cortical stroma; oMS: outer medullary stroma; iMS: inner medullary stroma; US: ureteric stroma; SCS: subcapsular stroma (red arrowhead). Black arrowheads indicate the lack of SPs in the transplanted organoids.

REVIEWERS' COMMENTS

Reviewer #2 (Remarks to the Author):

Authors have addressed all the concerns I brought up. The manuscript has much higher quality after the revision. I congratulate all authors!

Reviewer #3 (Remarks to the Author):

The review of the paper is satisfactory. I have no further comments

Point-by-point responses**Reviewer #2**

Authors have addressed all the concerns I brought up. The manuscript has much higher quality after the revision. I congratulate all authors!

Response

We appreciate these favorable comments.

Reviewer #3

The review of the paper is satisfactory. I have no further comments

Response

We appreciate these favorable comments.